# *Dubosiella newyorkensis* modulates immune tolerance in colitis via the L-lysine-activated AhR-IDO1-Kyn pathway

Yanan Zhang[1], Shuyu Tu[2], Xingwei Ji[1], Jianan Wu[3], Jinxin Meng[4], Jinsong Gao[1], Xian Shao[5], Shuai Shi[5], Gan Wang[1], Jingjing Qiu[6], Zhuobiao Zhang[1], Chengang Hua[1], Ziyi Zhang[1], Shuxian Chen ⓘ[1], Li Zhang[2] & Shu Jeffrey Zhu ⓘ[1,3,5] ✉

Commensal bacteria generate immensely diverse active metabolites to maintain gut homeostasis, however their fundamental role in establishing an immunotolerogenic microenvironment in the intestinal tract remains obscure. Here, we demonstrate that an understudied murine commensal bacterium, *Dubosiella newyorkensis*, and its human homologue *Clostridium innocuum*, have a probiotic immunomodulatory effect on dextran sulfate sodium-induced colitis using conventional, antibiotic-treated and germ-free mouse models. We identify an important role for the *D. newyorkensis* in rebalancing Treg/Th17 responses and ameliorating mucosal barrier injury by producing short-chain fatty acids, especially propionate and L-Lysine (Lys). We further show that Lys induces the immune tolerance ability of dendritic cells (DCs) by enhancing Trp catabolism towards the kynurenine (Kyn) pathway through activation of the metabolic enzyme indoleamine-2,3-dioxygenase 1 (IDO1) in an aryl hydrocarbon receptor (AhR)-dependent manner. This study identifies a previously unrecognized metabolic communication by which Lys-producing commensal bacteria exert their immunoregulatory capacity to establish a Treg-mediated immunosuppressive microenvironment by activating AhR-IDO1-Kyn metabolic circuitry in DCs. This metabolic circuit represents a potential therapeutic target for the treatment of inflammatory bowel diseases.

The human gastrointestinal tract harbors an enormous, diverse, and critical population of microorganisms (termed 'microbiota') that interacts intimately with its human host to form an interdependent and mutually restrained ecosystem, performing metabolic functions such as carbohydrate fermentation and vitamin biosynthesis[1,2]. The gut microbiota generates immensely diverse bioactive small-molecule metabolites that can be involved in signaling, mucosal barrier maintenance, or immune system modulation[3,4]. Given such a broad range of effects on host physiology and immunology, it remains unclear how specific microbes (and the small molecules they produce) interact to

[1]Department of Veterinary Medicine, College of Animal Sciences, Zhejiang University, Hangzhou, Zhejiang 310058, PR China. [2]Department of Cardiology, The First Affiliated Hospital of Guangdong Pharmaceutical University, Guangzhou, Guangdong 510080, PR China. [3]Department of Critical Care Medicine, Sir Run Run Shaw Hospital, Zhejiang University School of Medicine, Hangzhou, Zhejiang 310016, PR China. [4]College of Veterinary Medicine, Qingdao Agricultural University, Qingdao, Shandong 266109, PR China. [5]Shaoxing People's Hospital, Zhejiang University Shaoxing Hospital, Shaoxing, Zhejiang 312000, PR China. [6]College of Veterinary Medicine, Jilin Provincial Engineering Research Center of Animal Probiotics, Jilin Provincial Key Laboratory of Animal Microecology and Healthy Breeding, Jilin Agricultural University, Changchun 130118, PR China. ✉e-mail: shuzhu@zju.edu.cn

cause, sustain, mitigate, or predict gut-related diseases such as inflammatory bowel disease (IBD), ulcerative colitis (UC) and Crohn's disease (CD).

Excessive activation of Th1/Th17 and impairment of Foxp3+ regulatory T (Treg) cells have been described in the pathogenesis of IBD[5]. Treg-mediated immunosuppression via secretion of inhibitory cytokines of TGF-β and IL-10 is of great importance for inhibition of IBD[6]. They also preserve intestinal physiology by promoting epithelial barrier functions and tissue repair. Although specific intestinal microbes and related metabolites have been confirmed to be involved in the induction of Treg-mediated immune tolerance[7], the underlying molecular mechanisms are largely unknown.

The development of intestinal Tregs is affected by several metabolites derived from diet. Short-chain fatty acids (SCFAs), such as acetate, propionate, and butyrate are the main products of the fermentation of undigested carbohydrate in the gut, particularly in the colon. SCFAs induce accumulation of colonic Tregs (cTregs) through multiple mechanisms[6]. They can directly stimulate Treg proliferation through activation of GPR43 (also known as FFAR2)[8] and the differentiation of naïve CD4+ T cells into Treg cells through histone H3 acetylation of *Foxp3* by histone deacetylase (HDAC) inhibition[9].

Tryptophan (Trp) is an essential, aromatic amino acid that humans must acquire from diet. Dietary Trp can be metabolized to kynurenine (Kyn) by indoleamine-2,3-dioxygenase 1 (IDO1) activity in intestinal epithelial cells (IECs) and dendritic cells (DCs)[10,11] and contributes to the differentiation of naïve CD4+ T cells into Foxp3+ Tregs[12] through activation of the aryl hydrocarbon receptor (AhR)[13], a ligand-activated transcription factor and immune sensor[14] that has been implicated in IBD pathogenesis[15]. Although it has been demonstrated that the intestinal microbiome is important for Trp catabolism and AhR activation during IBD[16], the specific commensal species and the microbial metabolites that participate in activation of the AhR-IDO1-Kyn metabolic circuitry and their impact in establishing a Treg-mediated tolerogenic mucosal microenvironment has not been fully explored.

In this study, we identified an understudied, robust SCFA-producing commensal bacterium, *Dubosiella newyorkensis*, that can ameliorate dextran sulfate sodium (DSS)-induced colitis in mice by rebalancing Treg/Th17 responses and improving mucosal barrier integrity. We revealed that *D. newyorkensis* and its human homolog *Clostridium innocuum* (*C. innocuum*) generate L-Lys to modulate Treg-mediated immunosuppression by activating AhR-IDO1-Kyn circuits in mouse and human DCs. Collectively, these results point to a multi-factorial strategy used by *D. newyorkensis* and *C. innocuum* to maintain mouse and human gut immune homeostasis, respectively.

## Results

### Manipulation of the gut microbiota by single-antibiotic treatment restricts DSS-induced colitis

Wild-type C57BL/6J mice (WT) were given ampicillin (Amp), vancomycin (Van), neomycin (Neo), or metronidazole (Metro) either individually or as a cocktail containing all 4 (Abx) via oral gavage and subsequently exposed to DSS (Fig. 1a). Perturbation of the gut microbiota by both Abx and all 4 single-antibiotic treatments was confirmed as a significant reduction in 16S rDNA copies (Supplementary Fig. 1a). Neo treatment exerted the best effect in attenuating DSS-induced colitis, manifested by the least body weight loss (Supplementary Fig. 1b), longer colon (Fig. 1b, c) as well as lower colonic mRNA levels of proinflammatory cytokines (Fig. 1d) and IL-6 protein (Fig. 1e).

Next, we prepared fecal samples pooled from Neo mice and performed fecal microbiota transplantation (FMT) in microbiota-depleted Abx mice [Abx-FMT(N)] (Fig. 1a). As expected, FMT significantly restored the total fecal 16S rDNA copy number prior to DSS treatment, but not to the same level of vehicle (Veh) control (Supplementary Fig. 1c). Intriguingly, weight loss (Supplementary Fig. 1d), colon

shortening (Fig. 1c, f), histopathological lesions (Fig. 1g, h) and infiltration of IL-6-secreting macrophages (Fig. 1h) were greatly alleviated in the Abx-FMT(N), with results comparable to Neo group. These data suggest that the fecal microbiota of Neo mice may indeed contain specific intestinal microbes that can confer protection against DSS-induced colitis.

We carried out 16S rRNA gene sequencing and analysis on fecal samples collected from Veh, Abx, and single-antibiotic-treated mice at Day 0 (D0) and 7 (D7) post-DSS administration and found remarkable differences in microbiome composition based on antibiotic treatment, with or without DSS induction (Fig. 1i). There was a significantly increased relative abundance of *Bifidobacterium*, *Bacteroides*, *Akkermansia* and *Dubosiella* at D7, higher in the Neo-treated than other single-antibiotic-treated, and Abx mice (Fig. 1j, k).

### Colonization with *D. newyorkensis* mitigates DSS-induced colitis by restoring mucosal barrier function and inhibiting inflammatory responses

Among the bacterial taxa enriched by Neo treatment, *Bifidobacterium*, *Bacteroides*, and *Akkermansia* have already been reported to play protective roles during IBD[17–19], thus we focused on the *Dubosiella* genus (located within the family *Erysipelotrichaceae*) for the present study. We chose *D. newyorkensis* (abbreviated as Dub hereafter) isolated from a laboratory mouse[20] as the representative isolate to determine the role of genus *Dubosiella* in DSS-induced colitis. Antibiotic susceptibility testing of Dub confirmed its resistant to Neo and Metro but susceptibility to Amp and Van (Supplementary Fig. 2a).

Groups of WT mice were gavaged with Dub twice and then administrated DSS for colitis induction (Supplementary Fig. 2b). Since *Akkermansia muciniphila* (abbreviated as Akk hereafter) was also enriched after Neo treatment (Fig. 1j, k) and has been documented to be protective during IBD[21], it was chosen as a probiotic control in the study. *Enterococcus faecalis* (abbreviated as EF hereafter), which is a major cause of nosocomial infection, served as an unrelated Gram-positive commensal bacterial control. Indeed, Dub colonization exerted an even greater protective effect against DSS-induced colitis than Akk, with lower body weight loss (Supplementary Fig. 2c), longer colon length (Fig. 2a and Supplementary Fig. 2d), lower histopathological score (Fig. 2b, c) and less colonic macrophage infiltration (Fig. 2c). By contrast, EF colonization resulted in a colitic phenotype comparable to Veh controls (Fig. 2a–d and Supplementary Fig. 2c, d).

Furthermore, colonization with Dub led to better protection of the colonic mucosal barrier, as reflected by reduced apoptosis (Supplementary Fig. 2e), greater area of occludin-positive cells (Supplementary Fig. 2f), greater number of mucin-secreting goblet cells in the colonic epithelia and less disruption to the fine structure of the brush border and tight junctions (Fig. 2c). We also confirmed the differences of occludin, ZO-1 and Muc2 expression in the colonic intestinal epithelial cells (cIECs) (Fig. 2c, d and Supplementary Fig. 2g), while also showing that the expression of proinflammatory cytokines IL-1β, IL-6 and tumor necrosis factor-α (TNF-α) was significantly lower in the WT-Dub vs the WT-EF or WT-Veh, both in the colonic lamina propria (cLP) (Supplementary Fig. 2h) and peripheral blood (Supplementary Fig. 2i).

Additionally, WT mice gavaged with Dub for 5 consecutive days starting from day 3 post-DSS administration (Supplementary Fig. 2j) exhibited longer colon length (Supplementary Fig. 2k, l) and improved histopathology (Supplementary Fig. 2m, n) compared with mice receiving EF or untreated controls, suggesting that Dub also has therapeutic effects on DSS-induced colitis.

To further evaluate the probiotic effect of Dub on colitis, we used the mouse model of 2,4,6-trinitrobenzene sulfonic acid (TNBS)-induced colitis (Supplementary Fig. 3a). In this model, Dub colonization alleviated intestinal injury as indicated by longer colon (Supplementary Fig. 3b, c) and reduced histopathological lesions

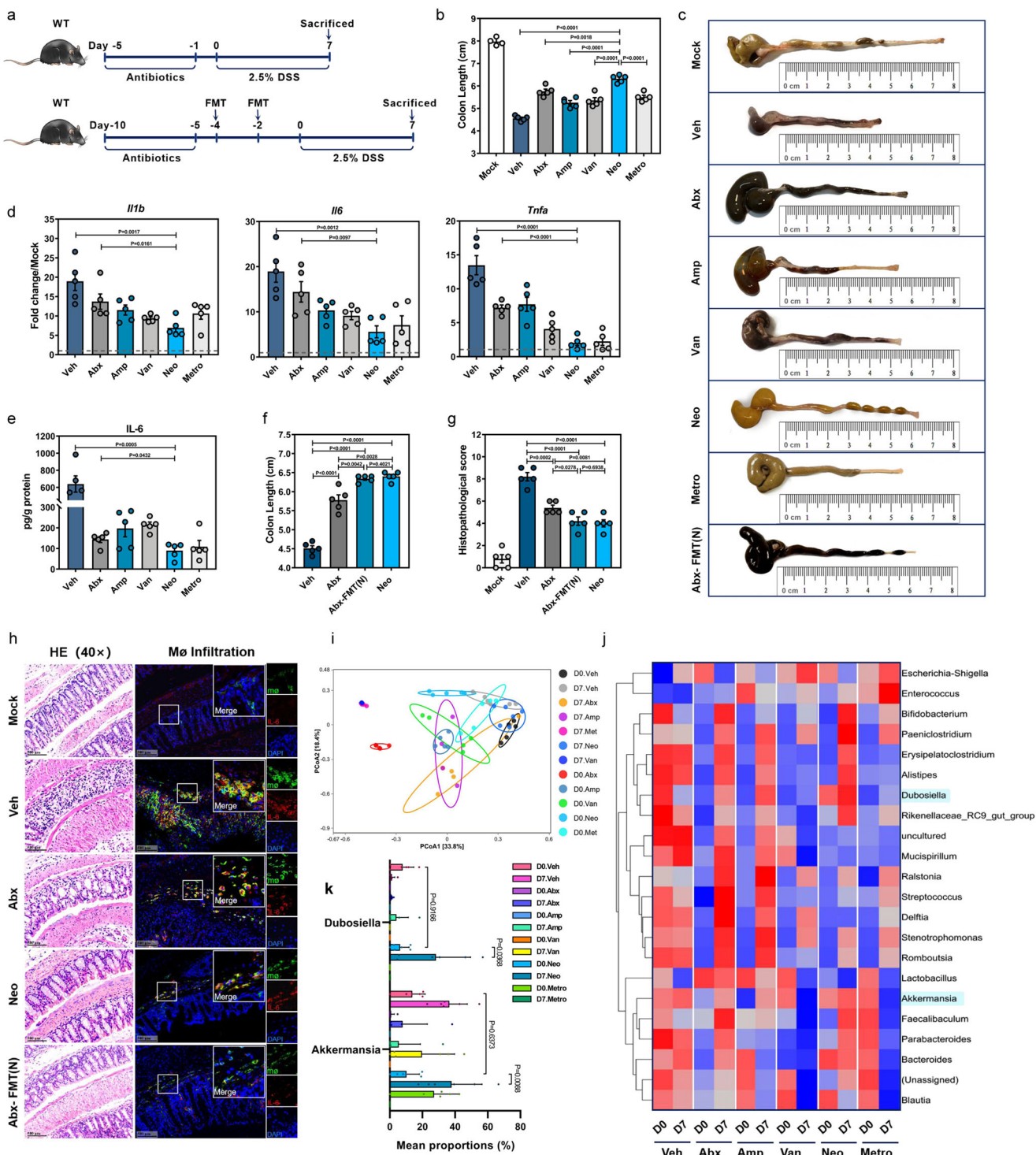

**Fig. 1 | Manipulation of gut microbiota affects mouse susceptibility to DSS-induced colitis. a** Outline of treatment regimens. **b**–**e** Wild-type C57BL/6J mice (WT, *n* = 4) received vehicle (Veh) or oral antibiotics (single Amp, Van, Neo, or Metro or in combination [Abx], *n* = 6) daily for 5 days and treated with 2.5% DSS in drinking water for 7 days. At day 7 (D7) post-DSS treatment, colon length (**b**, **c**), relative mRNA expression of proinflammatory cytokines *Il1b*, *Il6*, and *Tnfa* by qRT-PCR (**d**) and IL-6 protein level by ELISA (**e**) were determined in each group. **f**–**h** Abx mice were given fecal microbiota transplantation (FMT) from Neo-treated mice twice [Abx-FMT(N)] or Veh with a 48 h interval and then treated with DSS for 7 days (*n* = 5). WT B6 mice treated with Neo (Neo) were served as control. Colon length (**f**) was measured, histopathological changes were scored by HE staining (**g**, **h**) and

infiltration of IL-6-secreting macrophages (Mø) was determined by IFA (**h**). **i**–**j** Principal coordinate analysis (PCoA) (**i**) and heatmap of the relative abundance (**j**) of bacteria in fecal samples from mice treated with Abx or single antibiotics at D0 and D7 post-DSS administration. **k** Relative abundance of *Dubosiella* and *Akkermansia* at D0 and D7 among groups treated with different antibiotics (D7. Amp, D7. Van, *n* = 4; D0. Veh, D0. Abx, D0. Metro, D0. Amp, D0. Van, *n* = 5; D0. Neo, D7. Veh, D7. Neo, D7. Abx, D7. Metro, *n* = 6). Dashed lines at 1 indicate that the treatments have equal value as normalized controls. Results are representative of data generated in at least two independent experiments and are expressed as mean ± SEM, and 2-sided *P*-values were examined by the Student's *t*-test. Source data are provided as a Source Data file.

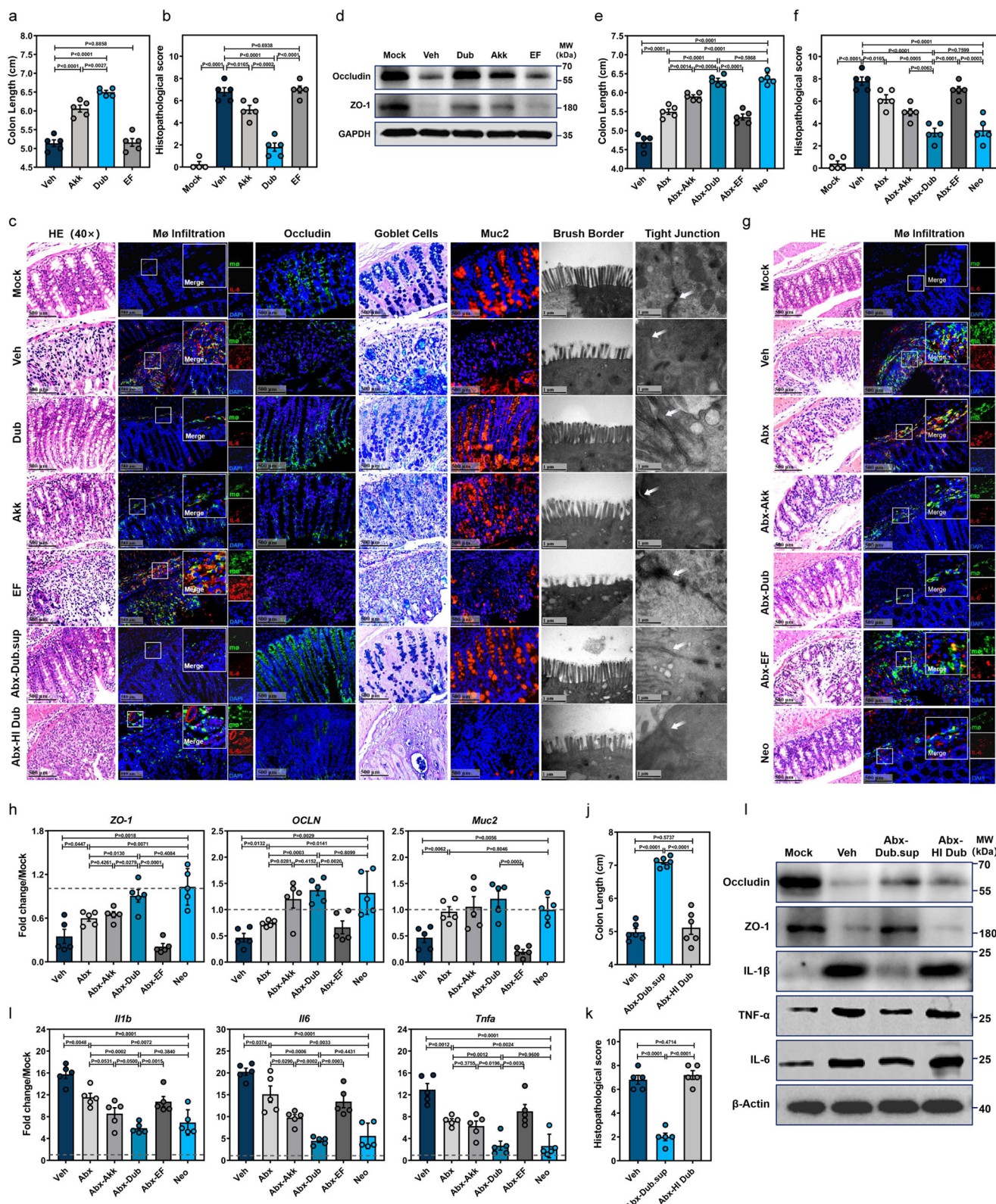

(Supplementary Fig. 3d, e). Dub colonization also increased the expression of *OCLN*, *ZO-1*, and *Muc2* in the cIECs (Supplementary Fig. 3f), and decreased the expression of *Il1b, Il6*, and *Tnfa* in the cLP (Supplementary Fig. 3g), in agreement with a role for Dub in protection of the mucosal barrier and prevention of intestinal inflammation.

To remove the effect of background microbes and link the probiotic effect directly to Dub colonization, Abx mice were gavaged with Dub (Abx-Dub), Akk (Abx-Akk) or EF (Abx-EF) and subsequently administrated with DSS (Supplementary Fig. 3h). All 3 bacterial isolates were able to repopulate the gut microbiota in Abx mice (Supplementary Fig. 3i). We observed less body weight loss (Supplementary Fig. 3k), longer colon length (Fig. 2e and Supplementary Fig. 3l), lower histopathological score (Fig. 2f, g) and less macrophage infiltration (Fig. 2g) in the colon of Abx-Dub compared with Abx-Akk, Abx-EF or Abx groups. These parameters were equivalent between Abx-Dub and Neo groups (Fig. 2e–g and Supplementary Fig. 3l). Similar to the Neo

**Fig. 2 | *D. newyorkensis* protects mice from DSS-induced colitis and attenuates resulting mucosal inflammation and barrier damages. a–d** Conventional wild-type C57BL/6J mice were colonized with $10^9$ CFU of *D. newyorkensis* (Dub), *A. muciniphila* (Akk), *E. faecalis* (EF) or vehicle (Veh) twice with 2-day break in between, then treated with DSS as previously described (*n* = 5). At D7 post-DSS administration, colon samples were collected to determine colon length (**a**), histopathological score (**b**) by HE staining, IL-6-secreting macrophage infiltration, Muc2 and Occludin expression by IFA, number of goblet cells by Alcian blue staining and microstructure of colonic epithelia by TEM (**c**). The expression of Occludin and ZO-1 in each group was determined by western blot (WB) (**d**). **e–i** Abx-treated WT mice were colonized with $10^9$ CFU Dub, Akk, EF, or Veh and exposed to DSS (*n* = 5). At D7 post-DSS treatment, colon length (**e**), histopathological score by HE staining (**f, g**), IL-6-secreting macrophage infiltration (**g**) were examined.

Meanwhile, the expression of *ZO-1*, *OCLN*, and *Muc2* in isolated colonic intestinal epithelial cells (**h**), and the expression of proinflammatory cytokines *Il1b*, *Il6*, and *Tnfa* in colonic lamina propria (cLP) (**i**) was detected by qRT-PCR.
**c, j–l** Conventional WT mice (*n* = 6) were orally administered heat-inactivated (HI) Dub, or Dub supernatant (Dub.sup) and exposed to DSS. At D7 post-DSS administration, colon length (**j**) was measured, histopathological changes were scored by HE staining (**c, k**, *n* = 5), level of tight junction proteins Occludin and ZO-1, and expression of proinflammatory cytokines IL-1β, IL-6, and TNF-α were determined by WB (**l**). Dashed lines at 1 indicate that the treatments have equal value as normalized controls. Results are representative of data generated in at least two independent experiments and are expressed as the mean ± SEM, and 2-sided *P*-values were examined by the Student's *t*-test. Source data are provided as a Source Data file.

controls, Abx-Dub had markedly increased expression of *ZO-1*, *OCLN* and *Muc2* in the cIECs (Fig. 2h) and significantly reduced of *Il1b, Il6*, and *Tnfa* in the cLP (Fig. 2i) compared with Abx-Akk, Abx-EF, Abx or Veh groups.

In order to better define the protective component of Dub, Abx mice were gavaged with filtered Dub supernatant (Dub.sup) or heat-inactivated (HI) Dub and then subjected to DSS treatment (Supplementary Fig. 3h). Interestingly, pretreatment with Dub.sup, but not HI Dub, ameliorated DSS-induced weight loss (Supplementary Fig. 3j) and disease phenotypes (Fig. 2j, k and Supplementary Fig. 3l), restored mucosal barrier functions (Fig. 2c, l), and inhibited inflammatory cytokine expression in the colon (Fig. 2l), even though the pasteurized Dub cells were detected in the feces of Abx mice before DSS administration (Supplementary Fig. 3i). Taken together, these results demonstrate a likely functional role for Dub-derived metabolites rather than direct activity of bacterial components in the attenuation of DSS-induced colitis.

## Dub prevents DSS-induced immunopathology by rebalancing CD25⁺Foxp3⁺ Treg and IL17⁺CD4⁺ T cell responses in the gut microenvironment

Using RNA-Seq analyses, we detected the downregulation of proinflammatory pathways, including those associated with leukocyte extravasation, NF-kB activation, and IL-17 signaling pathways, in colons from Dub mice compared with Veh controls at D7 post-DSS treatment (Fig. 3a and Supplementary Fig. 4a). Consistently, downregulation of proinflammatory *Il17* and *Ifng* and upregulation of anti-inflammatory *Il-10* and *Tgfb* in bulk cLP immune cells were observed in Dub or Akk, but not in EF groups (Supplementary Fig. 4b).

Given that Tregs play a central role in maintaining the immunosuppressive microenvironment in the gut via TGF-β and IL-10 secretion and an imbalance of Treg and Th1/Th17 responses is involved in the pathogenesis of IBD[5], we hypothesized that Dub colonization inhibits DSS-induced inflammation by inducing Foxp3⁺ Tregs and suppressing IFNγ⁺CD4⁺ T cells and IL17⁺CD4⁺ T cells. To test this, we colonized groups of Abx mice with Dub, administered DSS, and measured specific T cell subpopulations in the spleen, mesenteric lymph nodes (MLNs), and cLP by flow cytometry at D7 (Supplementary Figs. 2b and 3h). Respective gating strategies are presented in Supplementary Fig. 5. Since Akk can induce both colonic and peripheral Tregs in colitis[22], we included Akk as a positive control and EF as a negative control. Dub or Akk colonization significantly increased CD25⁺Foxp3⁺ Tregs and markedly decreased IL17⁺CD4⁺ T cells in the MLNs and cLP (Fig. 3b, c), but only mildly reduced IFNγ⁺CD4⁺ T cells in the cLP (Supplementary Fig. 4c) compared to EF or Veh controls, with or without gut microbiota at D7. A similar pattern was observed in the spleen, where CD25⁺Foxp3⁺ Tregs were significantly higher and IL17⁺CD4⁺ T cells were significantly lower in Dub or Abx-Dub groups compared with noncolonized controls at D7 (Supplementary Fig. 4d), indicating that Dub rebalanced Treg/Th17 response not only in the colon, but also in the peripheral blood in DSS-induced colitis. Notably,

Dub was superior to Akk in rebalancing Treg/Th17 response with or without an intact gut microbiota (Fig. 3b, c and Supplementary Fig. 4b–d).

Additionally, we gavaged conventional or Abx mice with Dub in the absence of DSS (Supplementary Fig. 4e) and found that CD25⁺Foxp3⁺ Tregs were also increased in the spleen and cLP, while IL17⁺CD4⁺ T cells were decreased in the MLNs and cLP without DSS induction (Supplementary Fig. 4f, g), implying that Dub contributes to the balance of Treg/Th17 response even under intestinal homeostasis.

Finally, WT or diphtheria toxin (DT)-treated Foxp3-DTR mice were gavaged with Dub or EF before DSS administration (Supplementary Fig. 4h). As expected, DT treatment severely compromised CD25⁺Foxp3⁺ Tregs in the spleen, MLNs and cLP in Foxp3-DTR mice (Supplementary Fig. 4i). Treg depletion in Foxp3-DTR mice reversed the differences observed between Veh-treated (WT-Veh) and Dub-colonized WT mice (WT-Dub) in disease phenotypes (Fig. 3d and Supplementary Fig. 4j), histological evaluation (Fig. 3e, f) and proinflammatory cytokine expression (Supplementary Fig. 4k), indicating that Foxp3⁺ Tregs are essential for the Dub protective effect against DSS-induced colitis. Interestingly, small differences still remained in the expression of *ZO-1*, *OCLN*, and *Muc2* between Veh-treated and Dub-colonized Foxp3-DTR mice (Supplementary Fig. 4l), suggesting that Dub might be able to promote mucosal healing independent of inducing Treg-mediated immunosuppression.

## Colonization with Dub prevents DSS-induced colitis by regulating CD25⁺Foxp3⁺ Tregs in part through the SCFA-GPR43 signaling axis

Given that Dub is a close genetic relative of *Faecalibaculum rodentium*[20], which is an efficient SCFA-producer that has been reported to be capable of protecting from intestinal tumor growth[23], we hypothesized that the protective phenotype of Dub might be due to robust SCFA production. Indeed, there were significantly higher concentrations of acetate, propionate (Prop), and butyrate in the fecal samples of Dub mice compared with EF groups or the Veh controls and the concentrations of Prop and butyrate were comparable between Dub- and Akk-colonized mice (Supplementary Fig. 6a). Consistent with this data, propionate and butyrate concentrations in Dub.sup were 10.9-fold and 2-fold higher than in Akk.sup, respectively. However, the concentration of acetate was 1.5-fold higher in Akk.sup than in Dub.sup (Fig. 3g).

Since propionate has the highest affinity for GPR43[24], which is required for the expansion and suppressive function of colonic Tregs (cTregs) in the DSS-induced injury[8,25], we expected that the differences in CD25⁺Foxp3⁺ Treg induction between WT-Dub and WT-Veh mice might be reproduced by Prop administration and both phenotypes might be lost in *Gpr43⁻/⁻* mice. Thus, we gavaged WT and *Gpr43⁻/⁻* mice with Dub twice, or supplemented with Prop in drinking water for 3 weeks, then administrated with DSS and assessed the frequencies of CD25⁺Foxp3⁺Tregs and IL17⁺CD4⁺ T cells in the spleen, MLNs and cLP (Supplementary Fig. 6b). Indeed, Prop administration led to its

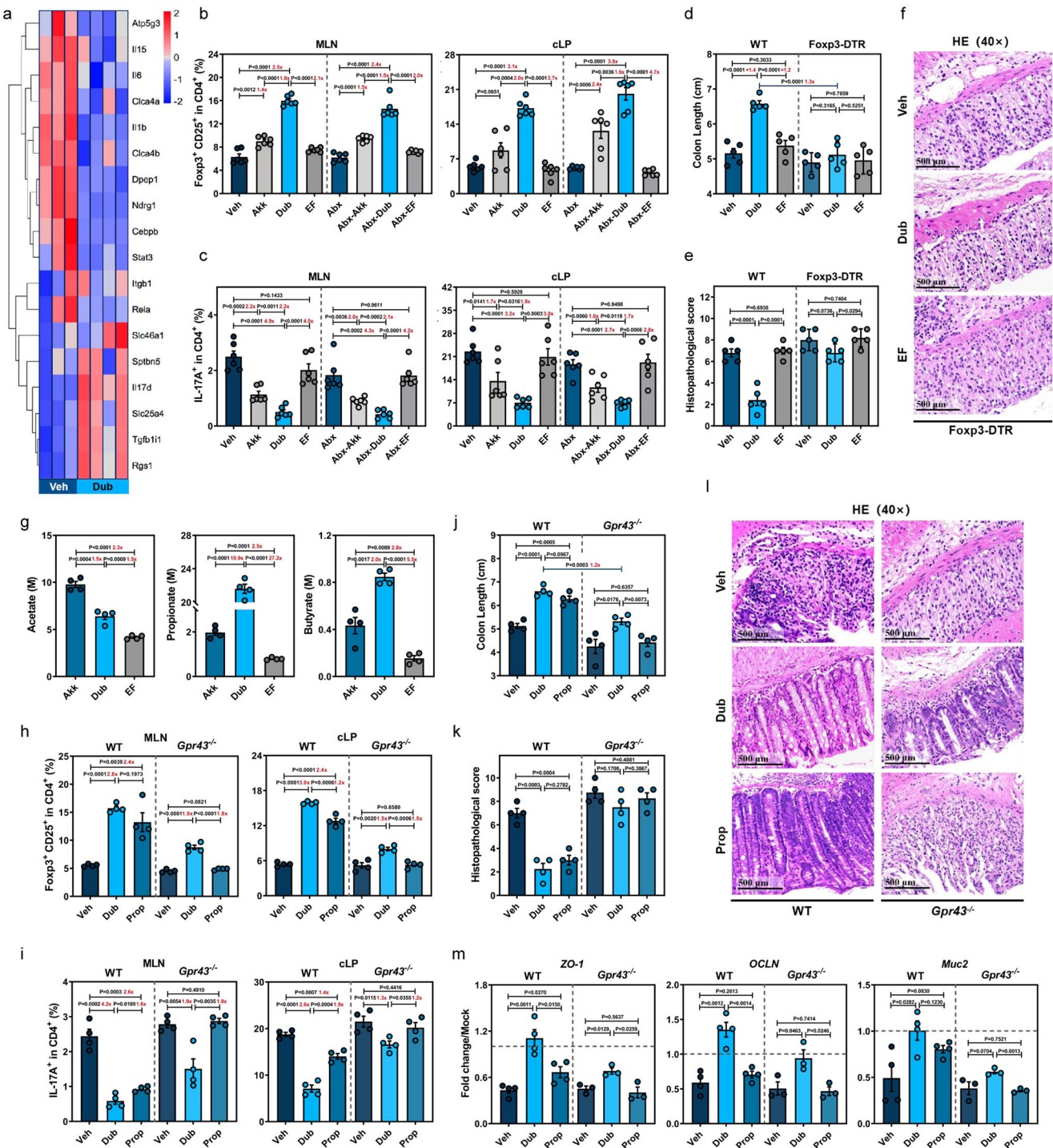

**Fig. 3 | *D. newyorkensis* prevents DSS-induced colitis by regulating CD25⁺Foxp3⁺ Tregs partially through the SCFA-GPR43 signaling axis. a** Heat map showing mRNA expression determined by bulk RNA-Seq in colon samples from vehicle (Veh)-treated ($n = 3$) or *D. newyorkensis* (Dub)-colonized wild-type C57BL/6J mice ($n = 4$) at D7 post-DSS treatment. **b, c** CD25⁺Foxp3⁺Tregs (**b**) and IL-17⁺ CD4⁺ T cells (**c**) in mesenteric lymph nodes (MLN) and colonic lamina propria (cLP) of conventional or Abx-treated WT mice colonized with Dub, *A. muciniphila* (Akk), or *E. faecalis* (EF) ($n = 6$) were examined by flow cytometry at D7 post-DSS treatment. **d–f** Conventional WT or Foxp3-DTR mice ($n = 5$) were colonized with 10⁹ CFU of Dub, Akk, or EF and subjected to DSS treatment. At D7 post-DSS administration, colon length measurement (**d**) and histopathological evaluation (**e, f**) were performed. **g** SCFA concentration in Dub, Akk, or EF culture supernatant determined by GC/MS ($n = 4$). **h–m** Conventional WT and *Gpr43⁻/⁻* mice ($n = 4$) were

colonized with Dub (10⁹ CFU) twice or administrated propionate (Prop) in drinking water for 3 weeks before DSS exposure. At D7 post-DSS administration, CD25⁺Foxp3⁺Tregs (**h**) and IL-17⁺ CD4⁺ T cells (**i**) in MLNs and cLP of conventional WT or *Gpr43⁻/⁻* mice ($n = 4$) were quantified by flow cytometry. Meanwhile, colon length (**j**), histopathological score by HE staining (**k, l**) ($n = 4$), and expression of *ZO-1*, *OCLN*, and *Muc2* in isolated colonic intestinal epithelial cells (cIECs) (WT, $n = 4$; *Gpr43⁻/⁻*, $n = 3$) (**m**) was determined. Fold of change in frequencies of CD25⁺Foxp3⁺Tregs or IL-17⁺ CD4⁺ T cells (**b, c, h, i**), colon length (**d**), or SCFA concentrations (**g**) between groups was calculated and presented with numbers in red. Dashed lines at 1 indicate that the treatments have equal value as normalized controls. Results are representative of data generated in at least two independent experiments and are expressed as mean ± SEM, and 2-sided *P*-values were examined by the Student's *t*-test. Source data are provided as a Source Data file.

significantly elevated concentration in the feces (Supplementary Fig. 6c) and exerted comparable effects in upregulating CD25+Foxp3+Tregs and downregulating IL17+CD4+ T cells in all 3 tissues compared with Dub colonization in WT mice. Unexpectedly, small differences in the frequencies of the 2 T cell subsets remained between *Gpr43*−/−-Dub and *Gpr43*−/−-Veh groups, whereas no differences were observed between *Gpr43*−/−-Prop or *Gpr43*−/−-Veh groups (Fig. 3h, i and Supplementary Fig. 6d), indicating a full involvement for GPR43 in the propionate-mediated Treg/Th17 rebalancing in the context of DSS-induced colitis.

Consistent with the results observed in Treg/Th17 regulation, a similar protective effect was exerted by WT-Prop compared with WT-Dub mice with respect to disease phenotype (Fig. 3j and Supplementary Fig. 6e, f), histological changes (Fig. 3k, l and Supplementary Fig. 6g), mucosal barrier integrity (Fig. 3m and Supplementary Fig. 6g–i) and inflammation (Supplementary Fig. 6j), while the protective phenotype was impaired in *Gpr43*−/− mice. However, despite a narrowing of the phenotype, there were still differences between the *Gpr43*−/−-Dub and *Gpr43*−/−-Veh groups, suggesting that in addition to SCFAs (especially propionate), other Dub-derived metabolites or GPR43-independent pathways might be involved in the Dub-driven induction of cTregs and amelioration of DSS-induced colitis.

## Dub induces CD25+Foxp3+ Tregs via enhancement of IDO1-mediated Trp metabolism in colonic dendritic cells

To search for other Dub-derived metabolites that might promote CD25+Foxp3+Treg induction, we performed untargeted metabolomic analysis on colon samples collected from conventional WT mice colonized with Dub, Akk, EF or Veh controls at D7, and a total of 253 compounds were detected. Partial least-squares discriminant analysis showed remarkably different metabolic profiles for Dub compared with the Veh controls (Supplementary Fig. 7a). Among the top 35 most abundant differential metabolites listed in the heatmap, 9 of them were statistically significantly differentiated between Dub-colonized mice and noncolonized control animals (Fig. 4a). KEGG analyses suggested that Trp metabolism was significantly lower in the DSS-induced group at D7 (Supplementary Fig. 7b), while colonization with Dub resulted in greater L-Kyn and quinolinic acid compared with the Akk, EF or Veh mice although the difference was not statistically significant (Fig. 4a).

To confirm the potential role of Kyn in rebalancing Treg/Th17 responses in intestinal inflammation settings, conventional WT and Foxp3-DTR mice were subjected to Kyn treatment prior to and throughout DSS administration (Supplementary Fig. 7c). Consistent with a previous publication[13], Kyn treatment resulted in a significantly higher level of CD25+Foxp3+Tregs and a drastically lower level of IL17+CD4+ T cells in the spleen, MLNs and cLP at D7 in WT mice (Fig. 4b, c and Supplementary Fig. 7d). Moreover, disease phenotypes (Fig. 4d and Supplementary Fig. 7e), histopathological changes (Fig. 4e, f) or inflammatory responses (Supplementary Fig. 7f) were the same in Foxp3-DTR mice, despite significant serum Kyn concentrations detected in both mouse lines at D7 (Supplementary Fig. 7g). Therefore, Dub colonization promotes the production of the Trp catabolite Kyn, which induces CD25+Foxp3+Tregs to exert a protective effect on DSS-induced colitis.

The majority of food-derived Trp is metabolized via the Kyn pathway through IDO1 in mucosal and immune cells or IDO2 and Trp 2,3-dioxygenase (TDO) in the liver[26] (Fig. 4g). Thus, we measured the kinetics of IDO1 expression in the colon of WT-Dub or WT-Veh at different times post-DSS treatment (Supplementary Fig. 7c) and determined that IDO1 was markedly elevated at D1, D3, and D5, but not on D7 in WT-Dub compared with WT-Veh controls (Fig. 4h and Supplementary Fig. 7h, i). In agreement with these results, Dub colonization led to a significantly increased serum Kyn concentration (Fig. 4i) at early phases of colitis progression (D0–D3).

To identify the cellular sources of IDO1 upregulation responding to Dub-derived metabolites, we tested *Ido1* expression in Abx-Dub and Abx-Dub.sup groups (Supplementary Fig. 7c) and demonstrated significantly increased *Ido1* expression in colonic lamina propria mononuclear cells (cLPMCs), but not in cIECs prior to and at D3 post-DSS administration (Fig. 4j). Moreover, adding Dub.sup to WT bone marrow-derived dendritic cells (BMDCs) resulted in significantly augmented *Ido1* expression compared to culture medium control, whereas EF.sup had no such enhancing effect (Fig. 4k); there was a similar trend for WT bone marrow-derived macrophages (BMDMs) under the treatment of Dub.sup although it was not statistically significant (Fig. 4k). On the contrary, *Ido1* expression was not altered in colonic organoids and slightly elevated in duodenal organoid at the presence of Dub.sup (Fig. 4k). These findings suggest that Dub-derived metabolites preferentially promote IDO1 expression in intestinal mononuclear phagocytes.

To exclude the possibility that the phenotypes were influenced by residual Abx-resistant commensal bacteria in our Abx group, germ-free (GF) mice were colonized with Dub (GF-Dub) or received FMT from conventional WT mice (GF-FMT) before DSS treatment (Supplementary Fig. 7j). Consistently, GF-Dub also had markedly higher *Ido1* levels in cLPMCs (Fig. 4l) and serum Kyn/Trp ratio (Fig. 4m) compared with GF-FMT. Correspondingly, GF-Dub displayed markedly higher CD25+Foxp3+ Tregs and remarkably lower levels of IL17+CD4+ T cells in the spleen, MLNs and cLP than GF-FMT (Fig. 4n, o and Supplementary Fig. 7k). Also, GF-Dub exhibited mitigated disease phenotype (Fig. 4p and Supplementary Fig. 7l), histological lesions (Fig. 4q, r) and inflammatory responses (Fig. 4s) compared with GF-FMT. Interestingly, Dub monocolonization resulted in higher *ZO-1*, comparable *OCLN*, and even lower *Muc2* level than in GF-FMT mice (Supplementary Fig. 7m). Altogether, we demonstrated that Trp metabolism in DCs was promoted by Dub, and the resulting Kyn enhancement from strengthened IDO1 expression rebalanced Treg/Th17 responses and ultimately exerted a protective effect on DSS-induced colitis.

## The Dub-derived metabolite L-Lys upregulates AhR-dependent IDO1 expression in colonic DCs

To investigate the bacterial metabolites mediating IDO1 upregulation, we performed untargeted metabolome analyses on Dub.sup after culture in chopped meat carbohydrate medium (CMC) (Fig. 5a). Liquid chromatography-tandem mass spectrometry (LC-MS/MS) analysis yielded 5 Dub-derived metabolites, N-acetyl-L-Asp (NAA), L-Lysine (Lys), L-Lactic acid (Lac), L-α-aminobutyric acid (Abu) and ketoleucine (4-methyl-2-oxopentanoic acid, 4-MOV) with relatively higher concentrations in both positive and negative ion-exchange chromatography (Supplementary Fig. 8a). These metabolites were also upregulated in the colon of Dub-colonized mice (Fig. 4a).

After pretreatment of WT mBMDCs with these 5 metabolites, we discovered that Lys and 4-MOV induced significantly higher *Ido1* (Fig. 5b). IFN-γ, a well-studied IDO1 stimulator in DCs, was used as a positive control[27]. In agreement with Gargaro et al.[28], we also determined that exogenous Kyn treatment of mBMDCs increased *Ido1* expression (Fig. 5b). Consistently, administration of Lys or 4-MOV to the WT mice (Supplementary Fig. 8b) significantly increased IDO1 expression in the colon at D3 (Fig. 5c). Among the 5 metabolites, Lys and Lac remarkably ameliorated colon shortening (Fig. 5d), tight junction damage and intestinal inflammation (Supplementary Fig. 8c), although the protective phenotype was not as robust overall compared with Dub.sup treatment.

Because 4-MOV showed no protective effect against DSS-induced colitis, we chose to focus on the potential role of Lys in modulating IDO1-driven Trp metabolism. Indeed, Lys pretreatment markedly elevated *Ido1* expression in WT mBMDCs in a dose- and time-dependent manner (Supplementary Fig. 8d, e). Furthermore, the addition of both Lys and Trp together to WT mBMDCs resulted in a significant decrease

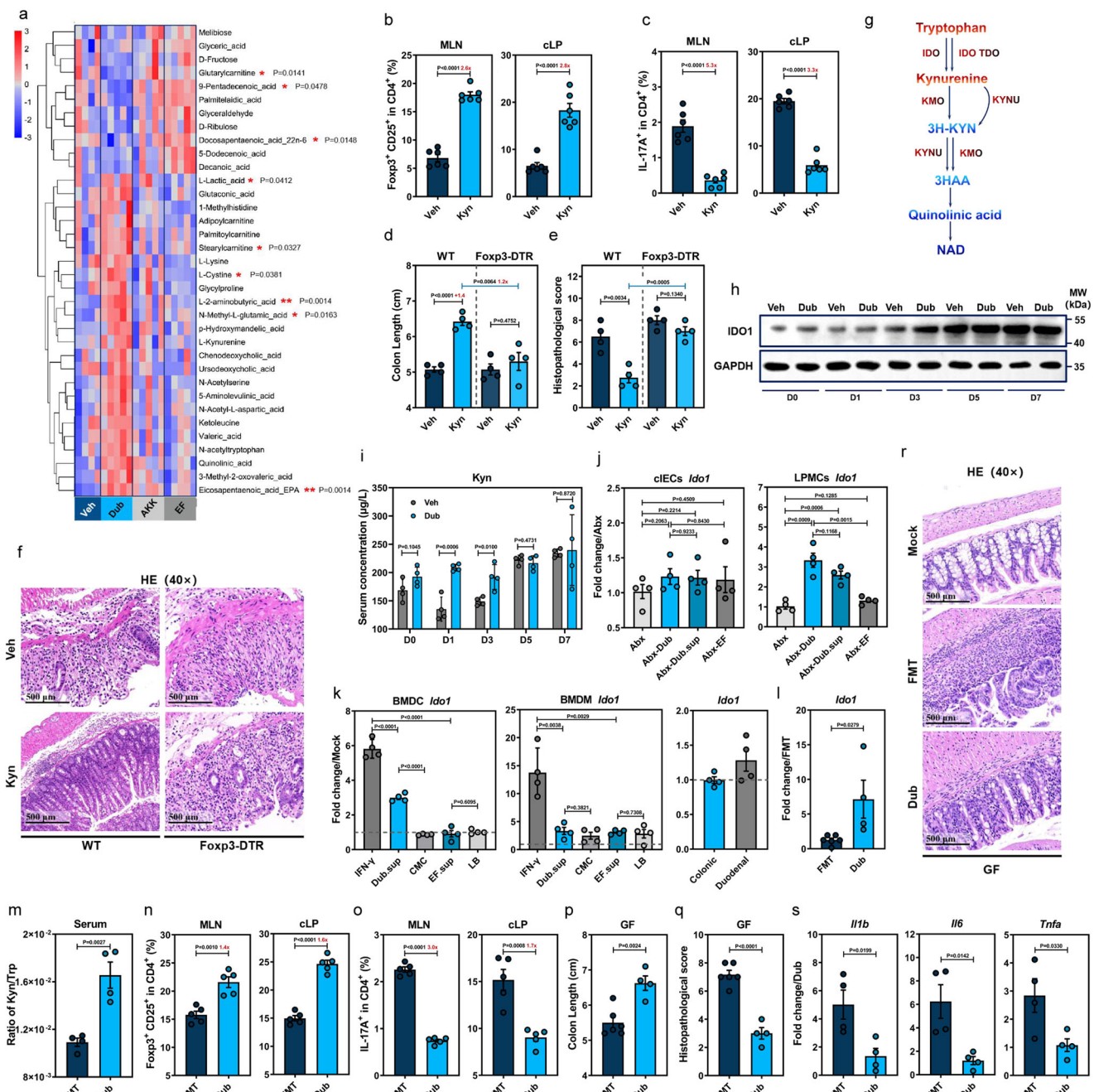

**Fig. 4 | *D. newyorkensis* promotes CD25+Foxp3+ Tregs through enhancement of IDO1-mediated Trp metabolism in dendritic cells. a** Heat map showing differential metabolites in colon at D7 post-DSS treatment from wild-type C57BL/6J mice (WT) treated with vehicle (Veh, *n* = 4) or colonized with *D. newyorkensis* (Dub), *A. muciniphila* (Akk), or *E. faecalis* (EF) (*n* = 5). Corresponding statistical annotations of differential metabolites represent the distinctions between Dub-colonized and noncolonized mice. **b**, **c** T cell subsets in mesenteric lymph nodes (MLN) and colonic lamina propria (cLP) from Veh- or Kyn-treated (10 mg/kg) conventional WT mice were examined by flow (*n* = 6). **d–f** WT or DT-treated Foxp3-DTR mice (*n* = 4) were treated with Kyn and subjected to DSS administration for 7 days, colon length (**d**) and histopathology (**e**, **f**) were examined. **g** Schematic of the Kyn catabolic pathway of Trp metabolism. **h**, **i** Kinetics of colonic IDO1 expression (**h**) and Kyn serum concentration (**i**) in Dub- or Veh-colonized conventional WT mice (*n* = 4) at indicated times post-DSS administration. **j** IDO1 expression in colonic intestinal epithelial cells (cIECs) or lamina propria mononuclear cells (cLPMCs) extracted from Abx-Dub, Abx-EF, Abx-Veh, or Abx-Dub.sup mice (*n* = 4) at D3 post-DSS administration. **k** *Ido1* expression in murine bone marrow-derived macrophages (mBMDMs) and bone marrow-derived dendritic cells (mBMDCs) or intestinal organoids treated with Dub or EF culture supernatants for 18 h. **l–s** Germ-free (GF) mice receiving fecal microbiota transplantation (FMT, *n* = 6) from conventional WT mice or colonized with Dub (*n* = 4) were exposed to DSS treatment for 7 days. *Ido1* expression in cLPMCs (**l**), serum concentration ratio of Kyn/Trp (*n* = 4) (**m**), T cell subsets (*n* = 5) (**n**, **o**) in MLN and cLP, colon length (**p**), pathological changes (**q**, **r**) (FMT, *n* = 6; Dub, *n* = 4) and proinflammatory cytokines in cLPMCs (*n* = 4) (**s**) were determined. Fold of change in frequencies of T cell subsets (**b**, **c**, **n**, **o**) or colon length (**d**) between groups was calculated and presented with numbers in red. Dashed lines at 1 indicate that the treatments have equal value as normalized controls. Results are representative of data generated in at least two independent experiments and are expressed as mean ± SEM, and 2-sided *P*-values were examined by the Student's *t*-test. Source data are provided as a Source Data file.

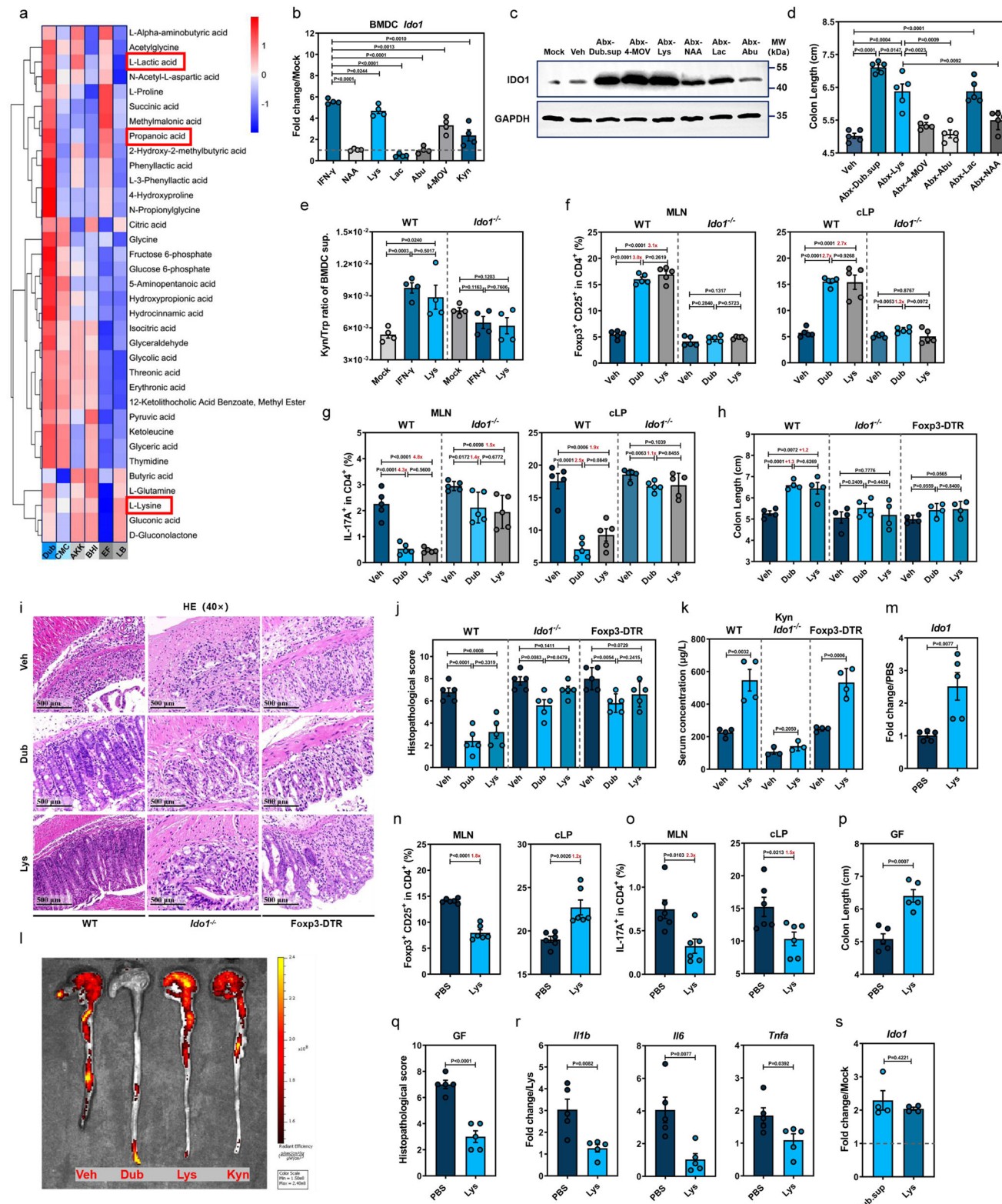

of Trp and remarkable increase of Kyn in the cellular supernatant (Supplementary Fig. 8f). This result was correlated with a more robust *Ido1* expression (Supplementary Fig. 8g) as shown by an increased Kyn/Trp ratio, which was lost in similarly treated mBMDCs collected from *Ido1⁻/⁻* mice (Fig. 5e), indicating that the Kyn pathway was enhanced by Lys in an IDO1-dependent manner.

Extracellular Lys is transported through SLC7A1, SLC7A2, and SLC7A3[29], while Trp is transported by system L, a heterodimer composed of a heavy chain (encoded by *Slc3a2*) and a catalytic L chain LAT1 (encoded by *Slc7a5*)[30]. Increased expression of *Slc7a1*, *Slc7a2*, *Slc7a5*, and *Slc3a2* was observed in mBMDCs collected from both conventional and GF mice and treated with Lys, compared with untreated cells (Supplementary Fig. 8h, i), while *Slc7a3* was not detected (data not shown). Thus, Lys-treated mBMDCs may upregulate SLC7A5 and SLC3A2 to increase Trp uptake, which fuels the Kyn pathway of Trp catabolism.

**Fig. 5 | L-Lys enhances Trp metabolism in dendritic cells to induce Treg-dependent immunosuppression by promoting IDO1 expression. a** Heat map showing differentially expressed metabolites of *D. newyorkensis* (Dub), *A. muciniphila* (Akk), *E. faecalis* (EF) and respective cultured supernatants (chopped meat carbohydrate broth, CMC; brain heart infusion medium, BHI; Luria–Bertani medium, LB) at 48 h. **b** *Ido1* expression in mouse bone marrow-derived dendritic cells (BMDCs) treated with N-acetyl-L-Asp (NAA; 1 mM), L-Lys (8 mM), L-lactic acid (Lac; 100 mM), L-α-aminobutyric acid (Abu; 0.1 mM), ketoleucine (4-MOV; 3 mM) and Kyn (0.2 mM) ($n = 4$). **c, d** IDO1 expression and colon length of Abx mice treated with metabolites were measured at D3 or D7 post-DSS administration, respectively ($n = 5$). **e** IDO1 activity in BMDCs treated with Lys or IFN-γ was determined by Kyn/Trp ratio of cell culture supernatant ($n = 4$). **f–j** Conventional WT, DT-treated Foxp3-DTR, or *Ido1*[−/−] mice were colonized with Dub or treated with Lys (20 mg/kg) and subjected to DSS administration for 7 days. T cell subsets (**f, g**) in mesenteric lymph nodes (MLN) and colonic lamina propria ($n = 5$) (cLP) colon length ($n = 4$) (**h**) and

histopathology ($n = 5$) (**i–j**) were determined. **k** Serum Kyn concentration of Lys-treated conventional WT, DT-treated Foxp3-DTR or *Ido1*[−/−] mice before DSS exposure (WT, Foxp3-DTR, $n = 4$; *Ido1*[−/−], $n = 3$). **l** In vivo imaging of colon from vehicle (Veh)-, Dub-, Lys- or Kyn-treated transgenic IL-17-EGFP mice. **m–r** Lys- or Veh-treated germ-free mice were exposed to DSS treatment for 7 days and colonic *Ido1* expression ($n = 5$) (**m**), T cell subsets ($n = 6$) in MLN and cLP (**n, o**), colon length (**p**), histopathology (**q**) and proinflammatory cytokines ($n = 5$) (**r**) were evaluated. **s** *Ido1* expression in Lys- (8 mM) or Dub supernatant (Dub.sup)-treated bone marrow-derived dendritic cells (BMDCs) extracted from GF mice at 18 h post-treatment ($n = 4$). Fold of change in frequencies of T cell subsets (**f, g, n, o**) or colon length (**h**) was calculated and presented with numbers in red. Dashed lines at 1 indicate that the treatments have equal value as normalized controls. Results are representative of data generated in three independent experiments and are expressed as mean ± SEM, and 2-sided *P*-values were examined by the Student's *t*-test. Source data are provided as a Source Data file.

To test the physiological relevance of the Lys-mediated IDO1 enhancement in suppressing inflammation via Treg induction in vivo, we gavaged conventional WT, *Ido1*[−/−] or Foxp3-DTR mice with Dub or Lys before DSS exposure (Supplementary Fig. 8j). It was found that Lys treatment augmented CD25[+]Foxp3[+]Tregs and diminished IL17[+]CD4[+] T cells to a similar extent that Dub colonization did in the spleen, MLNs, and cLP compared with Veh-treated WT mice, but Lys failed to modify CD25[+]Foxp3[+]Tregs in all 3 tissues or IL17[+]CD4[+] T cells in cLP of *Ido1*[−/−] mice (Fig. 5f, g and Supplementary Fig. 8k).

Moreover, Lys conferred clear beneficial effect on disease phenotypes (Fig. 5h and Supplementary Fig. 8l), histopathological changes (Fig. 5i, j), mucosal barrier function (Supplementary Fig. 8m), and inflammatory responses (Supplementary Fig. 8n) in conventional WT mice, whereas the protective effect was abolished in *Ido1*[−/−] or Foxp3-DTR mice. Consistent with the flow data, a mild protective effect remained in *Ido1*[−/−]-Dub and Foxp3-DTR-Dub mice as manifested by histopathological assessment (Fig. 5i, j). Furthermore, we determined that WT-Lys, but not *Ido1*[−/−]-Lys mice exhibited elevated serum Kyn concentration (Fig. 5k). Despite a smaller effect than Dub colonization, Lys and Kyn administration to DSS-treated IL-17-EGFP mice resulted in a similar downregulation in colonic IL-17 expression, as measured by in vivo imaging (Fig. 5l).

Subsequently, we determined whether the phenotype observed in conventional WT mice with normal gut flora was also true in GF settings. Indeed, GF-Lys displayed an increased serum Lys concentration prior to DSS exposure and an elevated fecal Lys concentration was confirmed in GF-Dub group (Supplementary Fig. 9b). As expected, Lys administration to GF mice (Supplementary Fig. 9a) caused higher *Ido1* expression in cLP (Fig. 5m), elevated serum Kyn concentration (Supplementary Fig. 9b), upregulated CD25[+]Foxp3[+]Tregs and downregulated IL17[+]CD4[+] T cells in the spleen and cLP (Fig. 5n, o and Supplementary Fig. 9c), but unexpectedly downregulated CD25[+]Foxp3[+]Tregs in the MLN (Fig. 5n). Lys also had a relieving effect on disease phenotype (Fig. 5p and Supplementary Fig. 9d), tissue injury (Fig. 5q and Supplementary Fig. 9e), mucosal barrier damage as well as local and peripheral inflammation (Fig. 5r and Supplementary Fig. 9f). Moreover, we pretreated GF mBMDCs with Lys and found significantly increased *Ido1* expression (Fig. 5s), implying that the stimulation of IDO1 in DCs by Lys was more likely due to cell-intrinsic changes not dependent on microbiota-derived signals.

For a better understanding of how Lys enhances *Ido1* expression in DCs, we performed RNA-Seq on Lys-treated WT BMDCs, revealing broad induction of AhR response genes (Fig. 6a). We confirmed the upregulation of *Ahr* (Fig. 6b, c) and the AhR target genes *Aldh1a3*, *Cyp1a*1, *Cyp1b1*, *Tiparp*, *Il1b*, and *Il6* in both conventional WT and GF mBMDCs at 6 h post-Lys induction (Supplementary Fig. 10a, b). Moreover, Lys led to translocation of AhR into the nucleus at 6 h post-treatment, implying an immediate effect of Lys on AhR (Fig. 6d). In accordance with this, western blot of Lys-treated mBMDCs showed

decreased cytoplasmic localization in parallel to increased nuclear AhR accumulation, comparable to that induced by the AhR agonist 6-formylindolo[3,2-b]carbazole (FICZ) (Fig. 6e). Activation of AhR and the resulting upregulation of *Ido1* expression in response to Lys were both inhibited by the AhR antagonist CH-223191 (Fig. 6f) or knockdown of *Ahr* (Fig. 6g and Supplementary Fig. 10c).

To evaluate the effects of Lys on AhR signaling in vivo, groups of WT mice were treated prior to DSS administration either with Lys in combination with CH-223191, or CH-223191 alone (Supplementary Fig. 10d). Consistent with the in vitro data, Lys administration did not boost *Ido1* expression in cLPMCs in the presence of CH-223191 at D3 post-DSS treatment (Fig. 6h). In agreement with the reported protective effects of AhR in DSS-induced colitis[31], WT-CH-223191 group displayed clearly impaired colonic *Ahr* expression (Supplementary Fig. 10e) and worsened disease phenotype (Fig. 6i, j). The differences in CD25[+]Foxp3[+]Tregs or IL17[+]CD4[+] T cells and proinflammatory cytokines in cLPMCs between WT-Lys and WT-Veh were lost in the presence of CH-223191 (Fig. 6k, l and Supplementary Fig. 10f, g). Collectively, these results suggest that Lys enhances IDO1 expression to confer protection against DSS-induced colitis via AhR activation.

## The human homolog of Dub, *C. innocuum*, produces L-Lys and promotes CD25[+]Foxp3[+] Tregs to alleviate DSS-induced injury by upregulating IDO1 expression in DCs

Recently, it has been documented that increases in the intestinal Erysipelotrichaceae family were causally related to a lower risk of IBD[32]. Thus, we performed a phylogenetic analysis of the Erysipelotrichaceae family and showed that an underrepresented human commensal species, *C. innocuum* (abbreviated as Clos hereafter), is a close relative to Dub (Fig. 7a, phylogenetic distance = 0.435). We compared the gut metagenomic datasets of the publicly available IBD cohorts and healthy matched controls[33–35], discovering that Clos was significantly decreased in patients with IBD, particularly those with UC, compared with healthy individuals (Fig. 7b). In addition, a USA cohort-based analysis displayed a lower abundance of Clos in patients with UC ($P = 0.0448$) or CD ($P = 0.1903$) in comparison with the healthy controls (Fig. 7c).

Similar to Dub, Clos generated Lys in the culture supernatant (Fig. 7d). Further, the Clos.sup and Lys both promoted *Ido1* expression in primary DCs extracted from human PBMCs (hPBMCs) (Fig. 7e) and WT mBMDCs (Fig. 7f), with increased Kyn/Trp ratio in mBMDC culture supernatant (Fig. 7g). In contrast, this phenotype was not observed in *Ido1*[−/−] mBMDCs (Fig. 7g), and was abrogated by CH-223191 (Fig. 7h) or AhR-specific small interfering RNA (siAhR) (Fig. 7i). In Lys-treated GF mBMDCs, upregulation of *Ahr* and its target genes, as well as Lys and Trp transporters were observed at 6 h post-treatment (Supplementary Fig. 11a, b). Similarly, higher *Ido1* expression was exhibited at 18 h post-treatment (Supplementary Fig. 11c). Together, these results indicate that Clos, the human homolog of Dub, may

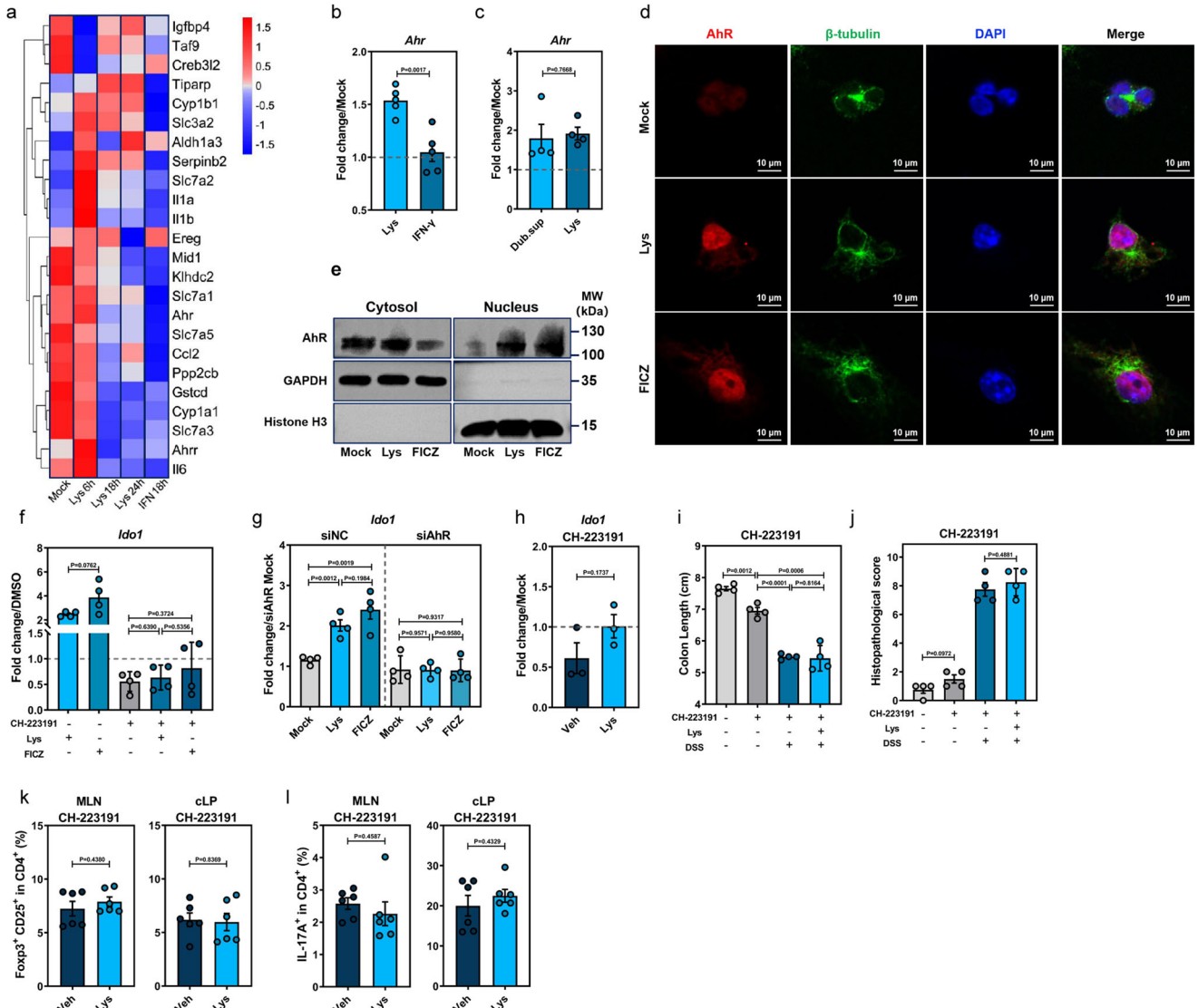

**Fig. 6 | L-Lys activates the AhR-IDO1 axis in DCs to induce immunosuppressive Treg responses. a** Transcriptomic analysis of mock-, Lys-, IFN-γ-treated mouse BMDCs at indicated times post-treatment (mock, $n = 3$; Lys 6 h, 18 h, 24 h, IFN 18 h, $n = 4$). **b** *Ahr* expression in Lys- or IFN-γ-treated BMDCs extracted from conventional WT mice ($n = 5$). **c** *Ahr* expression in Lys- or Dub.sup-treated BMDCs extracted from GF mice ($n = 4$). **d** Translocation of AhR (red) into the nucleus (DAPI, blue) from cytoplasm (β-tubulin, green) of mouse BMDCs after 6 h treatment with Lys or the AhR agonist 6-formylindolo[3,2-b]carbazole (FICZ). **e** AhR protein analysis in nuclear and cytoplasmic fractions of mock-, Lys-, or FICZ-treated mouse BMDCs at 6 h post-treatment by western blot. **f** Expression of *Ido1* in mouse BMDCs treated with Lys, FICZ, or the AhR antagonist CH-223191, or a combination of CH-223191 and Lys/FICZ ($n = 4$). **g** Expression of *Ido1* in mouse BMDCs transfected with negative control (siNC) or AhR-specific small interfering RNA (siAhR) and treated with Lys or FICZ at 18 h post-treatment ($n = 4$). For the experiments shown in (**a**–**g**)

above, the following were used: Lys (8 mM), FICZ (300 nM), and CH-223191 (10 µM). For experiments shown in (**h**–**l**) below, conventional WT mice ($n = 6$–8) were treated with Lys (20 mg/kg), CH-223191 (10 mg/kg), or Lys plus CH-223191 and subjected to DSS administration. **h** Expression of *Ido1* in the cLPMCs from Veh- or Lys-treated WT mice at D3 post-DSS administration in the presence of CH-223191 ($n = 3$). Colon length (**i**), histopathological changes by HE staining ($n = 4$) (**j**), CD25+Foxp3+Tregs (**k**) and IL-17+ CD4+ T cells ($n = 6$) (**l**) in MLN and cLP of Veh- or Lys-treated conventional WT mice in the presence of CH-223191 were examined at D7 post-DSS treatment. Dashed lines at 1 indicate that the treatments have equal value as normalized controls. Results are representative of data generated in three independent experiments and are expressed as mean ± SEM, and 2-sided *P*-values were examined by the Student's *t*-test. Source data are provided as a Source Data file.

also have an enhancing effect on Trp metabolism through the Kyn pathway by promoting IDO1 expression in both human and mouse DCs via AhR activation.

Next, we tested Clos or Clos.sup in WT mice and *Ido1*^−/−^ mice (Supplementary Fig. 11d), observing that both Clos colonization (Supplementary Fig. 11e) and Clos.sup administration significantly increased CD25+Foxp3+Tregs and decreased IL17+CD4+ T cells in the spleen, MLNs and cLP in WT mice, but not in *Ido1*^−/−^ mice (Fig. 7j, k and Supplementary Fig. 11f). Further, colonization of Clos or treatment with Clos.sup in WT mice mitigated colon shortening (Fig. 7l and

Supplementary Fig. 11g), colonic tissue injuries (Fig. 7m and Supplementary Fig. 11h) and inflammatory responses (Supplementary Fig. 11i), an effect which was not observed in *Ido1*^−/−^ mice. Thus, it is suggested that Clos may confer protection against colitis in humans by ameliorating unbalanced Treg/Th17 responses through enhancement of IDO1 expression.

## Discussion

A growing body of literature suggests that the intestinal microbiome plays a fundamental role in the maintenance of gut homeostasis

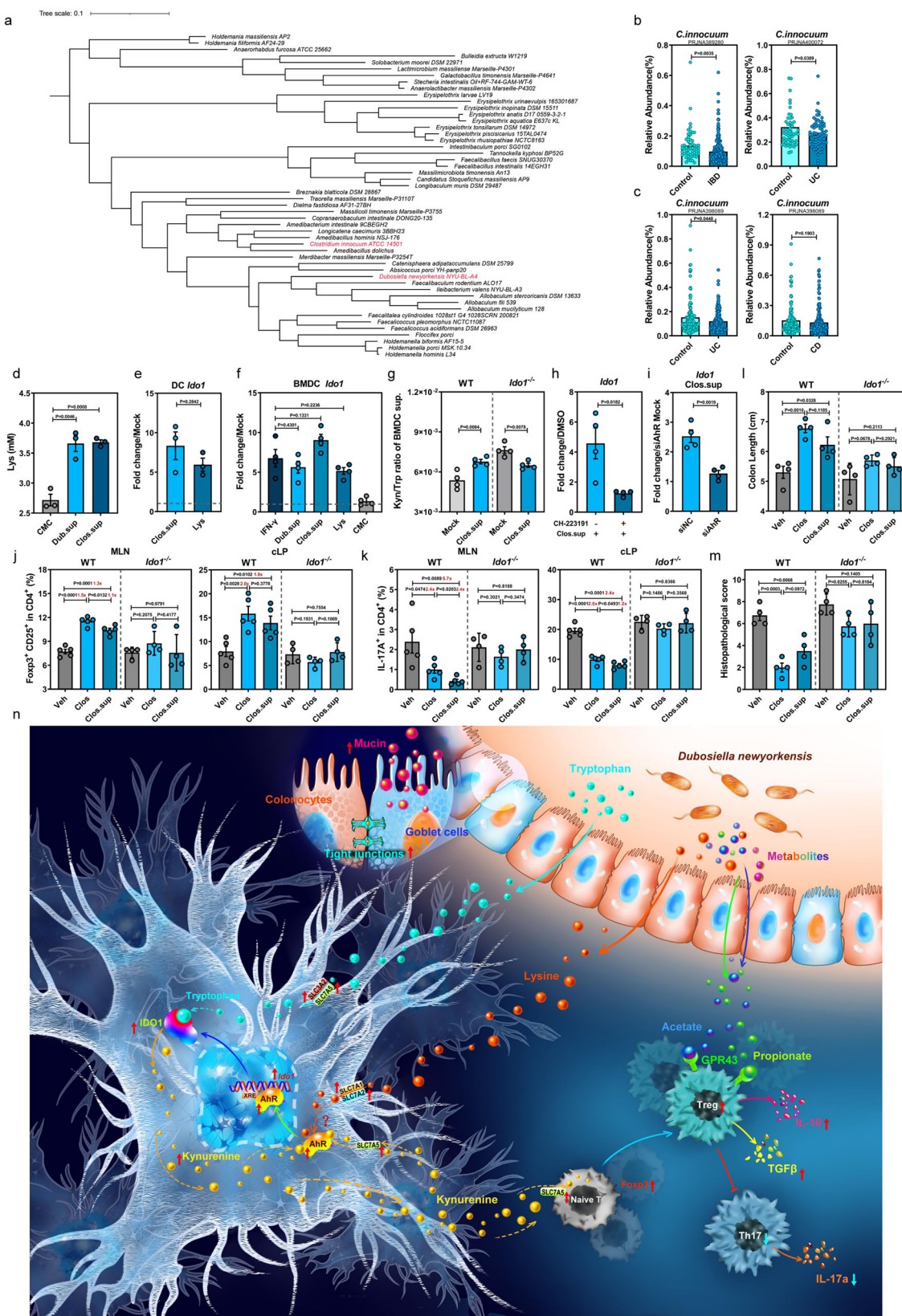

through enhancing epithelial barrier function and immune tolerance[4]. Metabolism has emerged as a new player in mucosal immune functions and inflammation with the advent of the new field of immunometabolism[36], though the role of specific microbiota-derived metabolites and their mode of action remain mostly elusive. Here we revealed that the Neo-resistant commensal bacterium, Dub, plays a

pivotal role in enhancing mucosal barrier integrity and rebalancing Treg/Th17 responses to maintain intestinal homeostasis through production of SCFAs (particularly propionate) and Lys (Fig. 7n). We have demonstrated a previously unknown role for intestinal microbe-derived Lys in promoting IDO1 expression in colonic DCs. By administrating Dub or Lys, we were able to skew endogenous Trp metabolism

**Fig. 7 | The human homolog of *D. newyorkensis*, *Clostridium innocuum*, generates L-Lys and protects against DSS-induced colitis by activating IDO1 expression in DCs. a** Phylogenic analysis of the Erysipelotrichaceae family and the *Clostridium innocuum* isolate. **b, c** Relative abundance of *C. innocuum* (Clos) in fecal samples from healthy individuals and patients with inflammatory bowel disease (IBD) (**b**), and specifically ulcerative colitis (UC) and Crohn's disease (CD) (**c**). Cohort PRJNA389280 (healthy, $n = 55$; IBD, $n = 221$), cohort PRJNA400072 (healthy, $n = 55$; IBD, $n = 73$), cohort PRJNA398089 (healthy, $n = 178$; UC, $n = 155$; CD, $n = 269$). **d** Levels of Lys in the culture supernatants of Dub (Dub.sup) or Clos (Clos.sup) ($n = 3$). **e** qPCR of *Ido1* mRNA in the Lys- or Clos.sup-treated human DCs extracted from peripheral blood mononuclear cells at 18 h post-treatment ($n = 3$). **f** qPCR of *Ido1* mRNA at 18 h post-treatment in wild-type C57BL/6J mouse (WT) bone marrow-derived dendritic cells (BMDCs) treated with 8 mM Lys, 20 ng/mL IFN-γ, Dub.sup, Clos.sup or chopped meat carbohydrate broth (CMC) ($n = 4$). **g** Kyn/Trp ratio of Clos.sup-treated, or nontreated BMDCs extracted from WT or *Ido1*$^{-/-}$ mice at 18 h post-treatment ($n = 4$). **h** Expression of *Ido1* in mouse BMDCs treated with Clos.sup,

CH-223191, or a combination of CH-223191 and Clos.sup ($n = 4$). **i** Expression of *Ido1* in mouse BMDCs transfected with negative control (siNC) or AhR-specific small interfering RNA (siAhR) and treated with Clos.sup at 18 h post-transfection ($n = 4$). **j–m** Conventional WT or *Ido1*$^{-/-}$ mice were colonized with $10^9$ CFU of Clos or treated with Clos.sup and subjected to DSS administration. At D7 post-DSS administration, CD25$^+$Foxp3$^+$Tregs (**j**) and IL-17$^+$ CD4$^+$ T cells (**k**) in mesenteric lymph nodes (MLN) and colonic lamina propria (cLP) (WT, $n = 5$; *Ido1*$^{-/-}$, $n = 4$), colon length (**l**) and histopathological changes by HE staining (**m**) were examined at D7 post-DSS treatment ($n = 4$). (**n**) Schematic illustration of the *D. newyorkensis* metabolite Lys protecting against DSS-induced colitis by rebalancing Treg/Th17 response through the activation of the AhR-IDO1-Kyn circuitry. Dashed lines at 1 indicate that the treatments have equal value as normalized controls. Results are representative of data generated in at least two independent experiments and are expressed as mean ± SEM, and 2-sided *P*-values were examined by the Student's *t*-test. Source data are provided as a Source Data file.

toward the Kyn pathway and rebalance Treg/Th17 responses to protect mice from colitis.

Multiple studies have demonstrated that antibiotic treatment can improve the symptoms of colitis, both in human trials[37,38] and DSS-induced mouse models[39,40], by shaping the composition of microbiota. Beneficial effect to clinical symptoms and inflammation suppression was linked to increased probiotic bacteria such as Bifidobacteria, *F. prausnitzii*[37] or *Enterobacter ludwigii* (*E. ludwigii*)[39]. In the present study, we found that Neo treatment could prevent DSS-induced colonic inflammation most effectively among all 4 broad-spectrum antibiotics (Amp, Van, Neo and Metro) (Fig. 1). This outcome does not resemble the results of a very recent study conducted by Li et al., also using multiple single-antibiotic treatments in a DSS-induced mouse model, in which Metro, but not Neo, was reported to induce DSS-colitis remission attributed to the expansion of *E. ludwigii*[39]. These discrepancies may be explained by differences in antibiotic regimens between the 2 studies (high-dose oral gavage in our study vs low-dose water distribution in Li et al.), as we have shown that these 2 treatment regimens could result in remarkably disparate susceptibility to caerulein-induced acute pancreatitis due to greatly distinct microbiota compositions[41].

The intestinal abundance of Dub has been reported to be associated with obesity[42] and Alzheimer's disease[43] in mouse models, however no functional studies have been performed to confirm its biological relevance in vivo. For the very first time, we comprehensively analyzed its potential role as a probiotic in both conventional and microbiota-depleted mouse models. Dub possesses a probiotic capacity superior to the well-known probiotic commensal bacterium Akk in improving colonic mucosal barrier damage and intestinal inflammation caused by DSS and TNBS (Fig. 2). Our findings that Dub relieves host mucosal and extraintestinal inflammation by rebalancing Treg/Th17 responses are perfectly in line with 2 landmark studies that demonstrate acetate and propionate regulate GPR43-dependent immunosuppression by cTregs under DSS or T cell-transfer conditions[8,25].

Although the key role of the gut microbiota in stimulating IDO1 activity has been demonstrated in both GF and Abx-treated mouse models[44], there has been very little focus on the modulatory influences on host Trp catabolism through the Kyn pathway exerted by specific commensal bacteria and microbial metabolites in IBD. Herein, we unraveled a novel role for Dub in enhancing host Trp metabolism through the Kyn pathway to regulate Treg/Th17 responses under steady-state or inflammatory conditions by promoting IDO1 expression in cLPMCs (Fig. 4). Further, we showed that Dub-derived Lys biased the function of DCs toward a regulatory and tolerogenic function to promote Tregs by upregulation of Kyn, through AhR-dependent IDO1 activity (Fig. 6). According to Gargaro et al. and others, both the 5' upstream region and internal noncoding region of *Ido1*

contains canonical xenobiotic responsive elements (XREs), which are AhR binding sites[28,45]. Additionally, we demonstrated that Lys upregulates SLC7A5 in DCs (Fig. S8h, i), which is a fundamental carrier for the IDO1-driven Kyn in establishing the AhR-IDO1-Kyn metabolic circuit in DCs via autocrine and paracrine signaling[28,46]. It needs to be noted that mild discrepancies still existed in Treg and Th17 between *Ido1*$^{-/-}$-Dub and *Ido1*$^{-/-}$-Veh/Lys groups (Fig. 5f, g), which could be plausibly explained by the fact that Dub is also able to rebalance Treg/Th17 responses through the SCFA-GPR43 axis (Fig. 3h, i). Thus, we are trying to generate *Gpr43*$^{-/-}$-*Ido1*$^{-/-}$ double knockout mice to confirm that these two immunomodulatory factors synergistically regulate Treg/Th17 responses in DSS-induced colitis.

Hardly any studies have shown that Lys impacts AhR activity, either directly or indirectly, and the molecular link between Lys catabolism and AhR activation remains uncertain, although it is possible that accumulation of crotonyl-CoA resulting from Lys catabolism reprograms IFN signaling[29], which may in turn activate the AhR-IDO1 axis[47]. Although it has been recently shown that different DC subsets acquire the IDO1 tolerogenic pathway and respond to immune-active Trp metabolites differently[28], further studies are needed to explore the regulatory role of Lys in the AhR-IDO1-Kyn circuitry in particular DC subsets. Experiments with cell type-specific conditional deletion of *Ahr* or *Ido1* are needed to address this concern. Lastly, we are making great efforts to generate a mutant Dub strain with *dapF* (encodes essential diaminopimelate epimerase for Lys biosynthesis) depletion, which will enable us to mechanistically dissect the specific role of Dub-synthesized Lys for mitigating DSS-induced colitis.

In summary, our comprehensive analysis demonstrates a previously unrevealed role of Dub in activating the AhR-IDO1-Kyn circuit in DCs, which improved the unbalanced Treg/Th17 response to ameliorate both mucosal and systemic inflammation in DSS-induced colitis. The finding that the human homolog Clos and its associated metabolite Lys can mitigate mucosal inflammation and improve mucosal healing in a mouse model may provide a basis for the development of microbiota-based therapeutic approaches for clinical IBD in humans.

## Methods
### Ethics statement
All animal experiments were strictly carried out in accordance with protocol no. 117113 approved by the Institutional Animal Care and Use Committee (IACUC) of Zhejiang University. Blood collection from healthy human donors was performed with the approval of the Ethical Committee of The First Affiliated Hospital of Guangdong Pharmaceutical University (Permit Number: 20210221). All participants provided written informed consent for sample collection and subsequent analyses.

## Bacterial strains

*A. muciniphila* (Akk) was obtained from American Type Culture Collection (ATCC, No. BAA835) and cultured in brain heart infusion medium (BHI; Oxoid) at 37 °C under anaerobic conditions. *D. newyorkensis* (Dab) and *C. innocuum* were purchased from the German collection of microorganisms (DSMZ, DSM103457 and DSM1286-1213-001) and cultured in chopped meat carbohydrate broth (CMC, Hopebio) at 37 °C under anaerobic conditions. *E. faecalis* (EF) was purchased from the China Center for Type Culture Collection (CCTCC, AB 2018154) and cultured in Lysogeny broth medium (Oxoid) at 37 °C. The concentration of each bacterial species was estimated based on the optical density at 600 nm ($OD_{600}$).

## Mouse strains

Six- to eight-week-old and sex-matched mice on C57BL/6 J background were used in this study. Germ-free (GF) C57BL/6J mice were bred and housed at the Shenzhen Gnotobio Biotechnology Co., Ltd. GF status was confirmed through 16S qPCR analysis before used for relative experiments, which were also carried out at Shenzhen Gnotobio Biotechnology Co., Ltd. Wild-type C57BL/6J (WT) mice were purchased from the Model Animal Research Center of Nanjing University (Nanjing, China). Foxp3-DTR mice were a gift from Dr. Bin Li from Department of Immunology and Microbiology, Shanghai Institute of Immunology, Shanghai Jiao Tong University School of Medicine. Foxp3-DTR mice were given intraperitoneal (i.p.) injections of 1 mg DTx (50 ng DT/g body weight) once per day for 7 consecutive days to guarantee successful depletion of Tregs. Splenocytes were analyzed to verify the elimination efficiency of Tregs by flow cytometry. *Indoleamine-2,3-dioxygenase 1* knockout mice ($Ido1^{-/-}$) were kindly provided by Dr. Yajing Wang from State Key Laboratory of Natural Medicines, Department of Physiology, China Pharmaceutical University. *G-protein coupled receptor 43* knockout mice ($Gpr43^{-/-}$) and IL-17-EGFP transgenic mice were purchased from Cyagen Biosciences Inc (Suzhou, China). The mice were routinely maintained in a specific-pathogen-free facility with a temperature- and humidity-controlled environment (22 ± 2 °C, 50 ± 10% humidity), and under a constant 12 h light/dark cycle, and were given free access to a regular chow diet (Gat# P1101F-25, Shanghai SLACOM) and water throughout study at Zhejiang University. All procedures were conducted in compliance with a protocol approved by the IACUC at Zhejiang University, China.

## Antibiotic treatment, fecal microbiota transplantation (FMT), and bacterial colonization

Mice were given ampicillin (Amp 10 mg), neomycin (Neo 10 mg), metronidazole (Metro 10 mg), or vancomycin (Van 10 mg), individually or in combination (referred to as Abx) daily for 5 days via oral gavage. Fecal samples collected from microbiota-depleted mice at the 5th day post-treatment were homogenized, plated on BHI agar with 10% sheep blood, and cultured under anaerobic conditions at 37 °C for 2 days followed by incubation under aerobic conditions at 37 °C for 1 day to confirm efficient microbial depletion.

For FMT experiments, 200 mg of pooled fecal pellets from WT mice were homogenized with sterile silica beads in 1.5 mL PBS at 45 Hz for 1 min and filtered with 70-μm strainers. Abx-treated WT mice or GF mice were subjected to gavage with 150 μL filtered stool homogenates (FMT experiments) or $10^9$ CFU of cultured bacteria (colonization experiments) twice with a 48-h interval. At 48 h after FMT or bacterial colonization, stool samples were collected to determine the efficiency of colonization prior to DSS administration for colitis model induction.

## DSS-induced colitis mouse model

Unless otherwise specified, WT or certain knockout mice were administered 2.5% DSS (Cat No: 60316ES76; Yeasen, shanghai, China) in drinking water for 7 days to establish the DSS-induced colitis mouse model.

To study the effect of antibiotics on mouse susceptibility to DSS-induced colitis, WT mice were pretreated with single antibiotics for 5 days, antibiotics administration was discontinued on day 6, and exposed to DSS treatment. Treated animals were weighed daily and fecal samples were collected at indicated times post-DSS treatment.

To evaluate the effect of bacterial strains on DSS-induced colitis, conventional WT mice were gavaged with $10^9$ CFU of *D. newyorkensis* (Dub), *A. muciniphila* (Akk), or *E. faecalis* (EF) twice with a 48-h interval, then subjected to DSS treatment. To evaluate the therapeutic effect of individual strains on experimental colitis without pretreatment, conventional WT mice were treated with 2.5% DSS in drinking water for 3 days, then gavaged from the 3rd day to the 8th day with $10^9$ CFU of Dub, Akk, or EF daily, maintaining 2.5% DSS in the drinking water.

To study the major effect of individual bacterial strains on DSS-induced colitis in the context of microbiota depletion, WT mice were pretreated with Abx for 5 days and on the 6th day gavaged with $10^9$ CFU of Dub, Akk, or EF twice with a 48-h break between treatments prior to DSS administration.

For pasteurization (HI) experiments, Abx-treated WT mice were orally gavaged for 5 days with $10^9$ CFU of Dub inactivated by pasteurization for 30 min at 70 °C, diluted in 150 μL PBS, and administered DSS on the 6th day.

Supernatants from cultured Dub or *C. innocuum* (Clos) were centrifuged at $6000 \times g$ for 10 min at 4 °C and then passed through polyether-sulfone filters (0.22 μm; Merck Millipore) to remove residual bacterial cells. Conventional or Abx-treated WT mice were gavaged with respective bacterial supernatants (200 μL per mouse) for 5 days and given DSS on the 6th day.

## TNBS-induced colitis mouse model

For presensitization, mice were shaved on the back just below the neck and painted with 1% TNBS (Meilunbio MB5523) mixed in an acetone/olive oil solution. Colitis was subsequently induced with 2.5% TNBS in 50% ethanol by rectal injection 7 days after presensitization as described previously[48].

## Animal protection study with metabolites

For the SCFA protection study, WT and $Gpr43^{-/-}$ mice were given 200 mM propionate in drinking water for 3 weeks and then administered 2.5% DSS. To test the effect of Dub-derived metabolites, mice were gavaged with L-lactic acid (Lac; 0.24 mg/kg), ketoleucine (4-MOV; 2.67 mg/kg), N-acetyl-L-Asp (NAA; 250 mg/kg), L-Lys (20 mg/kg), L-α-aminobutyric acid (Abu; 30 mg/kg) daily for 5 days before DSS treatment. For the Kyn protection study, WT or Foxp3-DTR mice were i.p. injected with 10 mg/kg Kyn every other day, 7 total doses from the 6th day previous to DSS to the 6th day post-DSS administration. For the Lys protection study, WT mice, Foxp3-DTR, or $Ido1^{-/-}$ mice were given L-Lys (20 mg/kg) via oral gavage once per day for 5 days before DSS administration. For AhR antagonist experiments, WT mice were treated i.p. with CH-223191 (10 mg/kg, MedChemExpress) daily for 10 consecutive days (from day −7 to day 2). During this period, mice were treated with 20 mg/kg Lys via oral gavage for 5 days (from day −5 to day 0) and then administered DSS.

## Fecal bacteria quantification

Fecal bacteria were quantified by qPCR (primers are listed in the table of Key Resources). Fecal bacterial DNA was isolated using a TIANamp Stool DNA Kit (TIANGEN), and qPCR was performed using SYBR Green Real-time PCR Master Mix (TOYOBO).

## DNA extraction, 16S rDNA amplicon sequencing, and data analyses

Fecal samples (~200 mg) were resuspended in Qiagen's ASL buffer and homogenized for 2 min. Total fecal DNA was extracted from the resulting supernatant using a QIAamp DNA Stool Mini Kit (Qiagen),

and DNA concentration and purity were measured by Qubit (Thermo Fisher Scientific). The stool DNA was then amplified using Phusion High-Fidelity PCR Master Mix (New England Biolabs) by PCR targeting the variable regions 3 and 4 (V3–V4) of the 16S rDNA. Multiplex sequencing of amplicons with sample-specific barcodes was performed using the MiSeq Illumina platform (Guangdong Magigene Biotechnology Co., Ltd.). Paired-end reads were merged into long sequences using FLASH v.1.2.7, a very fast and accurate analysis tool designed to merge paired-end reads when there are overlaps between reads1 and reads2. The merged sequences were then analyzed using the QIIME v.1.9.1 software package.

## Quantitative RT-PCR

Total RNA from bead-homogenized tissue samples or cell culture was extracted using TRIzol reagent (Invitrogen) following the manufacturer's instructions. PCR reactions were performed with HiScript II One Step qRT-PCR SYBR Green Kit (Vazyme) on a Gentier 96R Real-Time PCR System (TIANLONG, Xi'an, China). Transcript levels of the indicated genes were normalized to endogenous control GAPDH for each individual sample using the primers listed in the Key Resources table and quantified using the comparative critical threshold cycle $2^{-\Delta\Delta CT}$ method.

## Cytokine expression analyses

IL-1β, IL-6, and TNF-α protein levels in serum samples and colon homogenates were measured by the corresponding enzyme-linked immunosorbent assay (ELISA) kits (70-EK201B/3-96, 70-EK206/3-96, 70-EK282/4-96 MultiSciences) following the manufacturer's instructions.

## Western blot analysis

Tissues and cells treated as indicated were lysed with RIPA lysis buffer (Beyotime) and subjected to 10% SDS-polyacrylamide gel electrophoresis and then transferred to polyvinylidene difluoride membranes (Millipore). Proteins were further incubated with the indicated primary antibodies and then horseradish peroxidase-conjugated secondary antibodies. Protein bands were visualized using an enhanced chemiluminescence kit (Vazyme) with a ChemiDoc Touch Gel Imaging System (Bio-Rad).

## Tissue histology, immunostaining, and in vivo imaging

Colon 'Swiss rolls' soaked in 4% paraformaldehyde solution were dehydrated, embedded in paraffin, cut into 4-μm thick sections, and stained with haematoxylin and eosin (HE) using standard procedures. Slices were evaluated by an experienced pathologist in a blinded manner, and histological scores were assessed based on the following parameters according to previous research: inflammation, epithelial defects, crypt atrophy, dysplasia/neoplasia, and the area affected by dysplasia[49]. For the staining of goblet cells, colon sections were also stained in Alcian blue for 10–15 min and dehydrated in 100% alcohol and xylene.

For immunofluorescence (IFA) analysis, deparaffinized colon sections were blocked with 10% normal goat serum for 30 min at room temperature (RT). The slides were then incubated with specific primary antibodies at 4 °C for 12 h. Antibodies used: anti-IL-6 (1:50, 21865-1-AP Proteintech), anti-F4/80 (1:50, 28463-1-AP Proteintech), anti-occludin (1:100, Proteintech 13409-1-AP) and anti-Muc2 (1:100, 27675-1-AP Proteintech). Slides were then incubated with fluorescently labeled secondary antibodies (Jackson ImmunoResearch). Nuclei were stained with DAPI (Roche, Switzerland). TUNEL staining was performed using the In situ Cell Death Detection POD kit (Roche Diagnostics) on the Discovery XT according to the manufacturer's protocol. All analyses were performed using ImageJ software.

IL-17-EGFP transgenic mice were colonized with $10^9$ CFU Dub, treated with 10 mg/kg Kyn or 20 mg/kg Lys and subjected to 2.5% DSS administration as described above. At day 7 post-DSS administration, the colon was dissected from IL-17-EGFP transgenic mice in each group. In vivo imaging was performed using PerkinElmer (CLS136341/F) and representative images were taken.

## Transmission electron microscopy analysis

A 5-mm segment of fresh colon tissues was flushed with PBS and fixed in 2.5% glutaraldehyde at 4 °C for 4 h. After being rinsed in PBS, the tissue was further fixed in PBS containing 1% osmium tetroxide for 2 h at RT, rinsed in PBS, and dehydrated. The tissues were then embedded in Epon 812 overnight, then cured in an oven at 60 °C for 48 h. Sections of 80 nm thickness were cut on an ultramicrotome (RMC MTX) using a diamond knife. The sections were deposited on single-hole grids coated with Formvar and carbon and double-stained in aqueous solutions of 8% uranyl acetate for 25 min at 60 °C and lead citrate for 3 min at RT. Images were acquired on an H-7650 TEM (Hitachi, Ibaraki, Japan) at 80 kV of accelerating voltage. Regions of interest in the altered membranes were photographed by a Gatan 830 CCD camera (Gatan, CA, USA).

## Isolation of colonic intestinal epithelial cells and lamina propria mononuclear cells

Murine IECs and LPMCs were obtained from the colon as previously described[50]. Briefly, the colon was opened longitudinally and cut into pieces. After incubation with EDTA (5.5 mM) and dithiothreitol (DTT) (1 mM) in Hank's balanced salt solution (HBSS), vortexing and passing through a 70-μm cell strainer, the suspension of IECs was washed twice by centrifugation at $100 \times g$ for 2 min and collected for future experiments. The remaining lamina propria tissue was incubated with digestion solution containing collagenase (1 mg/mL) and DNase (0.2 mg/mL). The resulting LPMC cell suspension was subjected to Percoll-gradient separation and harvested for further experiments.

## Flow cytometry

For cell surface staining, single-cell suspensions were incubated on ice for 30 min with the following antibodies: FITC-conjugated anti-CD3 (11-0032-82, eBioscience, 1:150), eFluor 450-conjugated anti-CD4 (48-0041-82, eBioscience, 1:150), and APC-conjugated anti-CD25 (17-0251-82, eBioscience, 1:150).

For Treg cell analysis after staining of cells with CD3, CD4, and CD25 antibodies, lymphocyte suspensions were fixed and permeabilized using transcription factor buffer sets (562574, BD Pharmingen) according to the manufacturer's instructions and stained with anti-Foxp3-PE (12-5773-82, eBioscience, 1:75). For analysis of Th1 and Th17 cells, isolated tissue lymphocytes were stimulated for 5 h with cell stimulation cocktail plus protein transport inhibitors (00-4975-93, eBioscience). After incubation for 5 h, cells were washed in PBS and stained for cell death using fixable viability stain 570 (564995, BD Horizon, 1:100), FITC-anti-CD3, and eFluor 450-anti-CD4. Stained cells were fixed in fixation buffer (00-8222-49, eBioscience), permeabilized with intracellular staining permeabilization wash buffer (00-8333-56, eBioscience), and stained with anti-IL-17A conjugated to PerCP-Cyanine5.5 (45-7177-82, eBioscience, 1:75) and phycoerythrin-conjugated anti-IFN-γ (505808, Biolegend, 1:75).

## Isolation of murine BMDMs and BMDCs

Mouse bone marrow-derived macrophages (mBMDMs) were isolated from bone marrow extracted from mouse femurs and tibiae as previously described[51]. Cells were cultured at 37 °C and 5% CO₂ in DMEM supplemented with 20% FBS and 30% supernatant of filtered L929 cells, 100 U/mL penicillin, 100 μg/mL streptomycin at 37 °C and 5% CO₂ for 7 days prior to the experimental procedure. Murine bone marrow-derived dendritic cells (mBMDCs) were derived from bone marrow cells and stimulated with 20 ng/mL GM-CSF (Servicebio) and 10 ng/mL IL-4 (PeproTech) for 6 days. $5 \times 10^5$ of mBMDMs or mBMDCs were then

seeded into a 24-well plate and treated with L-lactic acid (100 mM), Kyn (0.2 mM), 4-MOV (3 mM), NAA (1 mM), L-Lys (8 mM), Abu (0.1 mM) for *Ido1* detection at 18 h post-treatment by qRT-PCR.

### Sorting of human dendritic cells

Healthy individuals admitted to the hospital for healthy examination were recruited as healthy donors in the study. Peripheral blood specimens were collected from seven healthy individuals (3 males and 4 females) aged from 22 to 41 and peripheral blood mononuclear cells (PBMCs) were extracted by density gradient centrifugation using human PBMC isolation kits (LTS10771 TBD) followed the manufacturer's protocol. After isolation, dendritic cells (DCs) were enriched from PBMCs using the EasySep™ Human Myeloid DC Enrichment Kit (19061 Stemcell), and enriched DCs were stained with BV421 anti-human CD11c (301628 BioLegend, 5 μL per sample). The enriched DCs were used for downstream assays. The protocol was reviewed and approved by the Committee for Ethical Review of The First Affiliated Hospital of Guangdong Pharmaceutical University, and written informed consent was obtained from each participant.

### Organoid culture

Murine duodenal and colonic organoids were cultured as previously described[52]. Briefly, crypts of duodenum were counted and embedded in 30 μL of Matrigel (Corning) at 10,000 crypts/mL and cultured in the mouse colonic organoid kit (K2204-MC, bioGenous). Surface area of organoids was quantified with ImageJ (National Institutes of Health). Colonic crypts were isolated and cultured as described previously[53]. Crypts of colon were counted and embedded in Matrigel and cultured in the mouse intestinal organoid kit (K2001-MI, bioGenous). At day 4 (duodenum) or day 3 (colon), organoids were stimulated with L-Lys for 18 h prior to RNA extraction and qRT-PCR detection.

### Organoid, BMDC, and BMDM treatment with bacterial culture supernatants

Bacterial supernatants were centrifuged at $6000 \times g$ for 10 min at 4 °C and then passed through polyether-sulfone filters (0.22 μm; Merck Millipore). Organoids, BMDCs, and BMDMs were incubated with bacterial culture supernatants (8%) diluted in respective cell culture medium for 18 h at 37 °C.

### IDO1 activity determination

IDO1 activity was measured in vitro as the ability to convert Trp into L-Kyn. Thus, 100 mg/L L-Trp was added to BMDCs in combination with 8 mM L-Lys or 20 ng/mL IFN-γ and incubated at 37 °C for 18 h. Cell culture SUPs were collected and concentrations of L-Kyn and L-Trp were measured by LC-MS/MS.

### AhR silencing and antagonization in BMDCs

2 μL Lipofectamine 2000 (Invitrogen 11668019) was added to 200 μL Opti-MEM (Gibco 31985070), gently mixed, and allowed to stand for 5 min. An siRNA pool specific for *Ahr* (1.25 μL) was mixed and allowed to stand for 20 min. BMDCs seeded in 24-well plates were incubated with the mixture for 10 min at RT and 500 μL of complete DMEM was added (Pricella PM150210). After 24 h, transfected mBMDCs were used for downstream assays. For inhibition experiments, BMDCs were pretreated with 10 μM CH-223191 (AhR antagonist) for 24 h and used for downstream assays.

### Detection of AhR translocation

mBMDCs were exposed to 8 mM Lys or 300 nM FICZ for 6 h. Cells were fixed with 4% paraformaldehyde in PBS for 15 min at RT, followed by a permeabilization step with PBS containing 0.5% Triton X-100 for 20 min at RT. Next, the Click reaction was performed by adding 100 μL of the reaction cocktail (Tublin 1:50, AF2835; AhR 1:100, AF6165 Beyotime) and incubated for 30 min at RT. Nuclei were stained with DAPI. Analysis was conducted using a confocal laser scanning microscopy (LSM880) with a 60x oil immersion objective. Tubulin was detected using the 488-nm laser, AhR was detected using the 594-nm laser, and DAPI was detected using the 405-nm laser. Images were collected using Zen software.

Protein was isolated from the 2 different cellular fractions using a Cytoplasmic and Nuclear Extraction Kit (R0050 Solarbio), following the manufacturer's instructions. Subsequently, samples were subjected to western blot analysis as described above. The cytoplasm-specific tubulin and nucleus-specific histone served as controls for appropriate fractionation. Antibodies used: anti-tubulin (1:50, AF2835 Beyotime), anti-histone (1:50, AF0009 Beyotime) and anti-AHR (1:100, AF6165 Beyotime).

### Transcriptomics analysis

BMDCs were pretreated with 8 mM Lys for 6, 18, or 24 h or 20 ng/mL IFN-γ for 18 h, then harvested for total RNA extraction with TRIzol reagent (Invitrogen). Samples were simultaneously assessed on an Agilent 4200 system (Agilent Technologies), Qubit 3.0 (Thermo Fisher Scientific), and Nanodrop One (Thermo Fisher Scientific). RNA-Seq libraries were generated and sequenced by Guangdong Magigene Biotechnology. Triplicate samples of all assays were constructed in an independent library, and the following sequencing and analysis were performed: Whole messenger RNA-Seq libraries were generated using a Next Ultra Nondirectional RNA Library Prep Kit for Illumina (New England Biolabs) following the manufacturer's recommendations. Clustering of the index-coded samples was performed on a cBot Cluster Generation System. After cluster generation, libraries were sequenced on an Illumina NovaSeq 6000 platform, and 150-bp paired-end reads were generated. Raw data in fastq format were processed by Trimmomatic (v.0.36) to acquire clean reads, which were mapped to NCBI Rfam databases to remove the rDNA sequences using Bowtie2 (v.2.33). Remaining mRNA sequences were mapped to the reference genome by Hisat2 (2.1.0). HTSeq-count (v.0.9.1) was used to obtain the read count and function information of each gene according to mapping results. Differentially expressed genes of 2 conditions/groups were determined using edgeR (v.3.16.5), and GO analysis was implemented using clusterProfiler (v.3.4.4), in which gene-length bias was corrected.

### Quasi-targeted metabolomics

Metabolites were extracted from the colon homogenates or bacterial SUP samples and sent to BGI (Shenzhen, China) for targeted metabolomics analyses. High-performance LC-MS/MS was used to perform high-sensitivity, wide-coverage, and high-throughput HM350-targeted quantification of 350 metabolites in these samples. In detail, appropriate amounts of experimental or quality control (QC) cell pellets were resuspended in 140 μL of a 50% water/methanol solution, lysed, and centrifuged, and the SUP was transferred to a new tube. A standard curve was prepared by serial dilution of an HM350 mixed standard. The experimental sample, QC sample, and standard were subjected to a derivatization reaction, and the resulting compounds were diluted in HM350 diluent and centrifuged at $12,000 \times g$ at 4 °C for 10 min. The supernatant was applied to LC-MS/MS analysis on an LC-MS QTRAP 6500+ (SCIEX). Chromatography was performed on a BEH C18 column (2.1 mm × 10 cm, 1.7 μm, Waters). Mass spectrometry was performed with an ESI+/ESI− source; further statistical and bioinformatics analyses were performed by BGI using routine procedures.

### LC-MS/MS determination of Trp and Kyn concentrations

Serum (100 μL) or BMDC culture supernatants were resuspended in 500 μL of acetonitrile/methanol (8:2) and centrifuged at $12,000 \times g$ for 20 min. The supernatant was then dried using a nitrogen blower. The precipitates were reconstituted in 100 μL of water/acetonitrile (8:2) containing 0.1% formic acid by thorough vortexing and centrifugation. Finally, the SUP (2 μL) was injected into the LC-MS/MS system for

analysis. A UHPLC–MS/MS system (ExionLC AD UHPLC-QTRAP 6500+, AB SCIEX Corp.) was used to quantitate Trp and Kyn at Novogene. LC-MS/MS was used to detect the concentration of a series of the standard solution. The concentration of the standard was used as the abscissa, and the ratio of the internal standard peak area was used as the ordinate to investigate the linearity of the standard solution.

## GC-MS/MS analysis of SCFA concentrations

20 mg of each fecal sample was placed in a 2 mL EP tube, and 1 mL of phosphoric acid (0.5% v/v) solution and a small steel ball were added. Samples were ground uniformly, then vortexed for 10 min and ultra-sonicated for 5 min. 100 μL of SUP was collected after the mixture was centrifuged at $10,000 \times g$ for 10 min at 4 °C. 0.5 mL MTBE (containing internal standard) solution was added, and the mixture was vortexed for 3 min and ultrasonicated for 5 min. After that, the mixture was centrifuged at $10,000 \times g$ for 10 min at 4 °C, then the SUP was collected and used for GC-MS/MS analysis.

For GC-MS/MS analysis of SCFAs, an Agilent 7890B gas chromatograph coupled to a 7000D mass spectrometer with a DB-FFAP column (30 m length × 0.25 mm i.d. × 0.25 μm film thickness, J&W Scientific, USA) was employed. Helium was used as carrier gas, at a flow rate of 1.2 mL/min. Injection was done in the split mode with an injection volume of 2 μL. The oven temperature was held at 90 °C for 1 min, then raised to 100 °C at a rate of 25 °C/min, then raised to 150 °C at a rate of 20 °C/min, held for 36 s, raised to 200 °C at a rate of 25 °C/min, held for 0.5 min and run for 3 min. All samples were analyzed in the multiple reaction monitoring mode; the injector inlet and transfer line temperatures were 200 °C and 230 °C, respectively. SCFA content was detected by MetWare (http://www.metware.cn/) based on the Agilent 7890B-7000D GC-MS/MS platform.

## Statistics and reproducibility

Statistical analyses were performed with Prism GraphPad software v.8.0. Error bars represent standard error of the means in all figures, and $P$-values were determined by 2-tailed Student's $t$-tests or ANOVA. $R^2$ was estimated for the correlation analysis of 2 continuous variables. A 2-sided $P$-value < 0.05 was considered statistically significant. All experiments were repeated, with the number of replicates stated in the figure legends. Representative images for western blots were from at least 3 independent sample preparations.

## Reporting summary

Further information on research design is available in the Nature Portfolio Reporting Summary linked to this article.

# Data availability

16S rRNA gene sequence data are available in the Sequence Read Archive (SRA) under BioProject accession PRJNA1013230. RNA-seq data are available in the SRA under BioProject accession PRJNA1012836. The mouse colon homogenates and bacterial cell culture supernatants metabolome data reported in this study have been deposited in the NGDC OMIX database (OMIX ID: OMIX005711; OMIX005712). All other data supporting the conclusions of this study are available in the paper and supplemental materials. Source data are provided with this paper.

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

## Acknowledgements

We appreciate Dr. Yajing Wang of China Pharmaceutical University and Dr. Bin Li of Shanghai Jiao Tong University School of Medicine for providing valuable *Ido1*$^{-/-}$ and Foxp3-DTR mice. We thank Dr. Lingdong Xu of Laboratory Animal Center, Zhejiang University for fundamental technical support with performing confocal microscopy. The professional editing service NB Revisions was used for technical preparation of the text prior to submission. This work is supported by grants from the National Natural Science Foundation of China (No. 32172864 and No. U21A20261) and the National Key Research and Development Plan of China (2022YFD1800804) to S.J.Z.

## Author contributions

S.J.Z. and Y.Z. designed the experiments. Y.Z., S.T., X.J., J.W., J.G., X.S., S.S., G.W., J.Q. ZB.Z., H.C. and ZY.Z. performed the experiments. Y.Z. and J.M. conducted bioinformatics analysis. S. S. and X.S. collected samples and data. S. C coordinated the project. L.Z. commented on and revised drafts of the manuscript. S.J.Z., Y.Z. and S.T. wrote the paper. S.J.Z. supervised research, coordination, and strategy.

## Competing interests

The authors declare no competing interests.
