## [Peer Review File · Nature Communications]

Dubosiella newyorkensis modulates immune tolerance in colitis through L-lysine-activated AhR-IDO1-Kyn metabolic circuitryREVIEWER COMMENTS

Reviewer #1 (Remarks to the Author):

It is well known that gut microbiota participates in the development of IBD through metabolic and immune regulation. In this study, the authors identified a murine commensal bacterium, *D. newyorkensis*, from alleviated colitis mice treated with antibiotics. On the one hand, *D. newyorkensis* was found to generate SCFAs especially propionate, which could ameliorate the colitis injury by promoting Treg in a GPR43-dependent manner. On the other hand, bacteria-derived Lys enhanced Treg-mediated immune suppression through AhR-IDO1-Kyn axis in DCs. Similar immune alterations and anti-colitis phenotypes were also observed in mice treated with clinical homologue of *D. newyorkensis*, i.e. *C. innocuum*. Overall, this manuscript reveals an interaction between commensal microbes and their hosts from a metabolic and immune perspective. However, there are several critical concerns about this paper.

Major Comments

1. Although the authors have fully explored the underlying mechanisms of immune suppression induced by Dub-derived metabolites, the reviewer is not particularly impressed by these findings as most key points have been reported in earlier research. These previously well-studied metabolites and downstream immune pathways, e.g. propionate-GPR43 axis in Tregs and AhR-IDO1-Kyn circuits in DCs, significantly weaken the originality of the paper. Mechanically, the authors did not provide much detailed information beyond known evidence. Thus, the existing data with limited innovation in this manuscript is currently insufficient to be published in Nature Communications. Novel bacterial metabolites with comprehensive molecular mechanisms would be of more interests.
2. Statistical methods need to be improved. One-way ANOVA with post-hoc analysis would be more appropriate for statistical comparison among multiple groups.
3. Keywords should be streamlined to retain the most core ones.
4. Introduction contains a large amount of information, which should be reorganized to make it more logical.
5. What is the dosage of Vancomycin?
6. Identification images of gene knockout mice must be provided.
7. In addition to the relative abundance of Bifidobacterium, Bacteroides, Akkermansia and Dubosiella higher in the Neo-treated than other single-antibiotic-treated, and Abx mice, there is also a significant increase in the abundance of Paeniclostridium. Why did the author initially overlook the changes in this bacterium?
8. Besides *D. newyorkensis*, are there any other strains that can be isolated and cultured, belonging to Dubosella genus? If so, have the authors attempted to verify the effects of other subspecies?
9. The results of transcriptome sequencing may sometimes be inconsistent with actual expression, and qRT-PCR should be used to verify the expression of key genes.
10. Why did the seemingly more valuable research on Cysteine and methionine metabolism in Figure S7B be abandoned in favor of studying tryptophan metabolism?
11. *D. newyorkensis* regulates Treg by SCFAs mediated GPR43 signaling or by Lys mediated AhR-IDO1-Kyn axis, furthermore, the relationship between the two pathways needs to be explained.

Specific Comments

Line 146-148: The authors state that Dub-treated mice exhibit less injury than Akk group, but no statistical difference between the two groups is shown in Figure S2K-N. Please explain this inconsistency.

Line 160-163: Information about the weight loss of Abx-Dub, Abx-Akk and Abx-EF mice is lacking in Figure S3J.

Line 186-188: Please add relevant references.

Line 226-227: According to Figure S6A, there is no statistical difference in butyrate or Prop levels between Dub and Akk mice. Please modify the inconsistent description in the text.

Line 279-281: Minor changes in protein levels, especially at D1 and D5 (Figure 4H), were not sufficient to make the claim that Dub treatment markedly increased IDO1. Better images plus quantitative scores would be needed to make this claim. Additionally, why does the IDO expression in Veh mice increase over time?

Line 299: Figure 4L showed the Ido1 levels in cLPMCs, but the figure legend in Line 619 claims that it is colonic expression.

Line 369-370: Figure 5J shows that Treg cells are decreased in MLNs from Lys mice, which is inconsistent with the content in the text.

Line 493-495: Please add relevant references.

Line 704: "Dub" is misspelled as "Dab".

Figure 1I: Similar colors make it difficult to distinguish D7 Neo and D0 Abx, D7 Met and D0 Neo. Consider using different symbols with the same color to represent D0 and D7 respectively.

Figure 5H: For the clearer presentation of data to the readers, authors should directly label the grouping information in this image.

Figure 5R, Figure S2E: Please provide scale bars.

Line 320: Fig. 4A should be changed to Fig. 5A.

All Figures and Supplementary Figures: In the qRT-PCR analysis, the data from normalized controls is recommended to be included in the images for an easy understanding.

Reviewer #2 (Remarks to the Author):

The authors investigated the role of commensal bacteria *Dubosiella newyorkensis* and its human equivalent, *Clostridium innocuum*, (as a last figure in the manuscript) in alleviating colitis in mice. These bacteria enhance immune tolerance by balancing Treg and Th17 responses and repairing mucosal injuries. The key mechanism involves the production of short-chain fatty acids, particularly propionate and L-Lysine. L-Lysine activates a novel metabolic circuit in dendritic cells, promoting tryptophan catabolism to kynurenine via indoleamine 2,3-dioxygenase 1 (IDO1) in an aryl hydrocarbon receptor (AhR)-dependent manner. This metabolic pathway offers a potential therapeutic target for inflammatory bowel diseases.

The study is a comprehensive yet concise exploration covering various facets, from microbial metabolism to immunoregulation. By employing multiple mouse models and delving into molecular mechanisms, it offers a complete understanding of the immunomodulatory role of these bacteria. Importantly, it

identifies a potential metabolic circuit as a potential therapeutic target.

As I said, I find the study to be largely comprehensive, with minimal room for additional commentary on my part. However, I do have a few targeted inquiries that, if addressed by the authors in the manuscript, could serve to further refine specific figures without altering the overarching message of the research. I will go through my suggestions according to the line number in the manuscript.

- Figure 1H: Is it possible to represent this data in a manner that is quantitatively measurable?
- It is difficult to understand how many times the authors repeated the experiments. I haven't pick up this information. Can you please add this into the figure legends? Moreover, it looks like it is only one experiment in many cases and if that's so, please change the SEM to standard deviation.
- Figure 1I: It looks like one ellipse is missing. Is that right? Or it is just color code issue that I cannot see it. Can you pick more distinguishable color on this panel?
- It will be nice to see how does microbiota look like after FMT in panel I-K in Figure 1?
- In line 121, only a single publication is cited to support the protective role of certain bacteria. Could you please broaden this section by including additional relevant studies? Providing a more extensive citation list would guide readers towards the appropriate literature for further investigation.
- In line 133, ... even greater probiotic effect than Akk... is written. what do you mean by probiotic effect? Explain a bit further.
- The authors have demonstrated that Akkermansia exhibits comparable effects in certain aspects of the study, as evidenced in Figure 2 and Supplementary Figure S2. Given the overlapping similarities in results, it would enhance the cohesiveness of the presentation to standardize the figures where Akkermansia data is included. For instance, how does the results look with Akkermansia when you perform TNBS model of colitis?
- Is the same set of genes upregulated in Akk treated group if one performed the experiment in Figure 3A?
- How does the phenotype look with monocolonization with Akkermansia?
- In Figure 4A, does the displayed data encompass all statistically significant findings? If so, could you please incorporate the corresponding statistical annotations? Additionally, could you elaborate on the distinctions between the various groups represented?

Reviewer #3 (Remarks to the Author):

In this manuscript, Zhang et al. discovered *D. newyorkensis*, an underappreciated murine commensal bacterium that produces short-chain fatty acids and can balance intestinal Treg/Th17 responses, thereby repairing mucosal barrier damage and mitigating DSS-induced colitis in mice. The authors demonstrated that *D. newyorkensis* and its human homologue *C. innocuum* produce L-Lysine to enhance tryptophan metabolism in human and mouse dendritic cells, which subsequently induced kynurenine production and Treg-mediated immunosuppression. Mechanistically, it appears that GPR43, IDO1, and the aryl hydrocarbon receptor were involved in *D. newyorkensis*'s immunoregulatory action.

Overall, the results of this study provide an important mechanistic understanding of how commensal bacteria create an immunosuppressive milieu through the secretion of immunomodulatory compounds. The mechanistic model builds with suitable validation and controls and is generally logical. The authors' ability to begin with a thorough functional analysis of a recently discovered murine commensal and end up with a human homolog that has a comparable biological effect is quite impressive. A couple of things are in need of more clarification or experimental evidence:

1. The authors have demonstrated that the relative abundance of the *Dubosiella* genus was significantly higher at D0 and D7 post-DSS treatment (Fig. 1K). How about the absolute abundance of *D. newyorkensis*? The authors need to ascertain whether this data aligns with the relative abundance of *Dubosiella*.
2. Please also provide the absolute abundance of *D. newyorkensis* in the fecal samples collected from mice that underwent fecal transplantation or antibiotic treatment (Fig. 1A).
3. Given that the bacterium produces high levels of SCFAs, particularly propionate (Fig. 3G and Fig. S6A), what pathway do the authors think *D. newyorkensis* uses to produce propionate?
4. Is there any experimental proof for the initial choice of propionate as the SCFA representative, without taking acetate or butyrate into account when examining *D. newyorkensis*'s immunosuppressive effects?

Minor points:

1. I've noticed that lactate is one of the five most abundant metabolites derived from the cultured supernatant of *D. newyorkensis*. Also, giving lactate to the animals receiving DSS treatment reduced intestinal inflammation, tight junction damage, and disease phenotypes (Fig. S8E). Why was L-lysine chosen as the target metabolite in the following study rather than lactate?
2. There was too much data included in Fig. 5. It may be helpful to split the figure into two for easier understanding.
3. Has the purity of the primary DCs isolated from human PBMCs been validated by the authors?
4. The color scheme of Fig. 5V should be matched to Fig. 5Y and 5Z.

REVIEWER COMMENTS

Reviewer #1 (Remarks to the Author):

It is well known that gut microbiota participates in the development of IBD through metabolic and immune regulation. In this study, the authors identified a murine commensal bacterium, *D. newyorkensis*, from alleviated colitis mice treated with antibiotics. On the one hand, *D. newyorkensis* was found to generate SCFAs especially propionate, which could ameliorate the colitis injury by promoting Treg in a GPR43-dependent manner. On the other hand, bacteria-derived Lys enhanced Treg-mediated immune suppression through AhR-IDO1-Kyn axis in DCs. Similar immune alterations and anti-colitis phenotypes were also observed in mice treated with clinical homologue of *D. newyorkensis*, i.e. *C. innocuum*. Overall, this manuscript reveals an interaction between commensal microbes and their hosts from a metabolic and immune perspective. However, there are several critical concerns about this paper.

Major Comments

1. Although the authors have fully explored the underlying mechanisms of immune suppression induced by Dub-derived metabolites, the reviewer is not particularly impressed by these findings as most key points have been reported in earlier research. These previously well-studied metabolites and downstream immune pathways, e.g. propionate-GPR43 axis in Tregs and AhR-IDO1-Kyn circuits in DCs, significantly weaken the originality of the paper. Mechanically, the authors did not provide much detailed

information beyond known evidence. Thus, the existing data with limited innovation in this manuscript is currently insufficient to be published in Nature Communications. Novel bacterial metabolites with comprehensive molecular mechanisms would be of more interests.

> We apologize if we did not appropriately highlight the novelty of our research in the first submission, thus giving the impression that our findings were not novel. We admit that SCFAs, especially propionate, have already been demonstrated to promote the differentiation of colonic Tregs and suppress the intestinal inflammatory responses via GPR43 activation (Smith et al. Science, 2013). However, we could not neglect this very important aspect, especially when we were trying to elucidate the probiotic function of an underappreciated commensal bacterium in a more comprehensive way. *D. newyorkensis*, but not its SCFA metabolite propionate, could still induce immunosuppressive colonic Treg (cTreg) responses and ameliorate DSS-induced colitis to some extent in *Gpr43*^{-/-} mice (Fig. 3H-M). It was this unexpected phenotype that inspired us to identify a previously unrevealed role for L-lysine (Lys) in modulation of cTreg responses by enhancing kynurenine (Kyn) production via Trp catabolism in dendritic cells (DCs). In brief, our study is the first to describe a role for Lys as an indoleamine 2,3-dioxygenase 1 (IDO1) enhancer, skewing Trp metabolism towards the Kyn pathway in DCs to promote Treg-mediated immunosuppression in a DSS-induced colitis mouse model. This study is quite novel on several fronts: First, the majority of studies on the immunomodulatory effect of commensal bacteria-produced Trp metabolites focus on indole derivatives, some of which are widely acknowledged as AhR ligands. Nevertheless, there has been very little

focus on the regulatory role of intestinal microbes in host Trp metabolism via participation of the Kyn pathway in maintenance of gut immune homeostasis by Treg induction. Second, our finding that Lys, an essential amino acid that does not possess an aryl group, can activate AhR in DCs is quite surprising and will be the focus of future studies. Third, our study highlights the importance of *D. newyorkensis*'s human homologue, *C. innocuum*, in mitigating mucosal inflammation and improving mucosal healing in the DSS-induced colitis mouse model, which may provide a basis for the development of microbiota-based therapeutic approaches for clinical IBD in humans.

2. Statistical methods need to be improved. One-way ANOVA with post-hoc analysis would be more appropriate for statistical comparison among multiple groups.

> We now have reanalyzed the data of relative weight loss between multiple groups using one-way ANOVA with post-hoc analysis, and rectified the description in the text and Figure Legends as the reviewer suggested.

3. Keywords should be streamlined to retain the most core ones.

> We now have streamlined the keywords and retained 6 core ones according to the reviewer's suggestion.

4. Introduction contains a large amount of information, which should be reorganized to make it more logical.

> We now have reorganized the "Introduction" section for the reviewer's consideration.

The changes in the text are shown with yellow highlighting (Line 53-77).

5. What is the dosage of Vancomycin?

> We have added the missing information to the text (it was 10 mg).

6. Identification images of gene knockout mice must be provided.

> We now have provided the identification images of KO mouse strain genotyping as following to address the reviewer's concern.

A. *Ido1*^{-/-} mice genotyping result:

B. *Gpr43*^{-/-} mice genotyping result:

C. Foxp3-DTR mice genotyping result:

Figure 1. Gel images of KO mouse strain genotyping. (A) *Ido1*^{-/-} mice; (B) *Gpr43*^{-/-} mice; (C) Foxp3-DTR mice.

7. In addition to the relative abundance of Bifidobacterium, Bacteroides, Akkermansia and Dubosiella higher in the Neo-treated than other single-antibiotic-treated, and Abx mice, there is also a significant increase in the abundance of *Paeniclostridium*. Why did the author initially overlook the changes in this bacterium?

> We really appreciate that the reviewer has pointed this out. We did not pick *Paeniclostridium* as a target because we were looking for a potential probiotic commensal bacterium, and one of two species belonging to this genus, *Paeniclostridium sordellii* (*P. sordellii*), is a pathogen that causes rapidly fatal infections characterized by severe edema, extreme leukemoid reaction and lack of innate immune response (French et al., Anaerobe, 2022). Another recent study conducted by Bernard et al. demonstrated that human infections caused by this toxin-producing and spore-forming anaerobic bacterium are associated with a treatment-refractory toxic shock syndrome (Bernard et al., PLoS Pathogens, 2022). We believe the relative abundance of *Paeniclostridium* increased

because they are more resistant to neomycin (Neo) and became relatively prosperous by occupying ecological niches and utilizing resources (nutrients or territory) when competitors had been wiped out by Neo.

8. Besides *D. newyorkensis*, are there any other strains that can be isolated and cultured, belonging to *Dubosiella* genus? If so, have the authors attempted to verify the effects of other subspecies?

> As far as we know, *D. newyorkensis* was the only strain belonging to *Dubosiella* genus that can be isolated and cultured and is commercially available (NYU-BL-A4; ATCC TSD-64) other than *Dubosiella muris* (*D. muris*), which was recently (late 2022) isolated from the caecal/colon content of an $APC^{min/+}$ $Msh^{2-/-}$ mouse (Afrizal et al., Cell Host & Microbe, 2022). It shares 91.10% of 16S rRNA gene sequence homology with *D. newyorkensis* and its genome has been assigned as a previously unknown species within the genus *Dubosiella*. However, this newly identified strain was inaccessible at the time, thus we were unable to use it in our research, unfortunately.

9. The results of transcriptome sequencing may sometimes be inconsistent with actual expression, and qRT-PCR should be used to verify the expression of key genes.

> We completely agree with the reviewer that the transcriptomic sequencing results could be inconsistent with the actual expression level of genes. Thus, we did verify the expression of our target genes with qRT-PCR. For instance, we detected the level of *Il1b*, *Il6*, *Il17a* and *Tgfb* expression in bulk cLP immune cells by qRT-PCR (Fig. S4B and Fig.

S4K), which could be used as verification for the transcriptomic analysis represented in Fig. 3A. But to address the reviewer's concern, we now have provided verification of the expression of key genes including *Rela*, *Stat3* and *Cebpb* as following (Fig. 2 in this PBP letter). These data are in line with the results of the transcriptome sequencing shown in Fig. 3A.

For the RNA-Seq analysis on Lys-treated WT bone marrow-derived dendritic cells (BMDCs) at various time points post treatment, we used qRT-PCR to validate the expression of *Ahr* and *Ahr* target genes including *Aldh1a3*, *Cyp1a1*, *Cyp1b1*, *Tiparp*, *Il1b* and *Il6* with qRT-PCR (Fig. S10A). The results were consistent with the heatmap of transcriptomic analysis shown in Fig. 6A.

Figure 2. The expression of *Stat3*, *Rela* and *Cebpb* in the colonic lamina propria cells (cLPs) of *D. newyorkensis* (Dub)-colonized (10^9 CFU) or noncolonized mice at day 7 (D7) post-DSS treatment as determined by qRT-PCR.

10. Why did the seemingly more valuable research on Cysteine and methionine metabolism in Figure S7B be abandoned in favor of studying tryptophan metabolism?

> Indeed, KEGG analyses suggested that Cys and Met metabolism was significantly

higher in the *D. newyorkensis*-colonized mice vs Veh-treated controls (Fig. S7B). In our study, we chose the metabolite candidates with relatively higher concentrations that appeared in the untargeted metabolomic analyses of both colon samples from colonized mice at D7 and the supernatant of *D. newyorkensis* cultures. L-cystine, the differential metabolite that was significantly enriched in the pathway of Cys and Met metabolism in vivo (Fig. 4A), was not observed in the untargeted metabolomic analysis of the supernatant of *D. newyorkensis* cultures in vitro (Fig. 5A), suggesting that *D. newyorkensis* does not produce L-cystine and thus increases L-cystine in the colon of colonized animals via an indirect route. Therefore, we focused on Trp metabolism and tried to corroborate the function of Lys in mitigating IBD instead of choosing L-cystine and Cys and Met metabolism as our target in the present study.

Actually, we did notice the phenotype and subjected groups of wild-type B6 mice to oral gavage with L-cystine prior to DSS administration. Interestingly, L-cystine pretreatment led to greatly alleviated disease phenotypes, pathological changes and expression of proinflammatory cytokines and significantly improved expression of tight junction proteins compared with untreated control animals (Fig. 3 in this PBP letter). Because we are trying to discern a role for L-cystine in mitigating DSS-induced colitis in an upcoming paper, we did not include these data in the current manuscript. But to satisfy the reviewer's concern, we've provided these unpublished results in the responses letter as following:

Figure 3. Oral administration of L-cystine reduces susceptibility to DSS-induced colitis. Groups of conventional wild-type C57BL/6J mice (WT; n=4-6) were gavaged with L-cystine (200 mg/kg) or vehicle (Veh) for 5 days before DSS administration. Colon length (A and B), histopathological score by H.E. staining (C), expression of IL6 and IL1b in cLPs (D) and expression of ZO-1 and OCLN in cIECs (E) were determined.

11. *D. newyorkensis* regulates Treg by SCFAs mediated GPR43 signaling or by Lys mediated AhR-IDO1-Kyn axis, furthermore, the relationship between the two pathways needs to be explained.

> We performed the following studies to address the reviewer's concern about the relationship between the two pathways of propionate-GPR43 signaling and Lys-mediated AhR-IDO-Kyn metabolic circuitry:

We harvested BMDCs from WT or *Gpr43*^{-/-} mice, treated with Lys and detected the

expression of *Ahr* and the *AhR* target genes including *Cyp1a1*, *Cyp1b1*, *Tiparp* and *Aldh1a3* at 6 h post-Lys induction (Fig. 4A in this PBP letter). We determined equivalent upregulation of *Ahr*, *Tiparp* and *Aldh1a3* after Lys pretreatment between WT and *Gpr43*^{-/-} BMDCs, whereas the Lys-induced upregulation of *Cyp1a1* and *Cyp1b1* was impaired with the absence of GPR43 in DCs. Moreover, Lys led to comparable expression level of *Slc7a1*, *Slc7a2*, *Slc7a5* and *Slc3a2* between WT and *Gpr43*^{-/-} BMDCs (Fig. 4B in this PBP letter), implying that stimulation of AhR in DCs by Lys seemed not to increase the uptake of Lys or Trp.

To test whether propionate can still induce Tregs in the absence of IDO1 *in vitro*, splenic naïve (CD44^{lo} CD62L^{hi}) CD4⁺ T cells were cultured in the presence of T-cell antigen receptor and CD28 signaling plus TGF-β with or without 0.1 mM propionate for 96 h (Furusawa et al., Nature, 2013). Indeed, propionate significantly increased the frequency of Foxp3⁺ cells in the *Ido1*^{-/-} splenic CD4⁺ T cells to a similar extent as that of the WT group (Fig. 4C in this PBP letter), indicating that propionate exerts a promoting effect on Treg induction independent of IDO1.

To further confirm what we observed *in vitro*, we administered 150 mM propionate to WT, *Gpr43*^{-/-} and *Ido1*^{-/-} mice for 3 weeks, or with 20 mg/kg Lys for 5 days as described in our paper, then administered DSS and assessed the frequencies of CD25⁺Foxp3⁺Tregs and IL17⁺CD4⁺ T cells in the cLP, MLN and spleen. Interestingly, we observed significantly increased CD25⁺Foxp3⁺Tregs and greatly decreased IL17⁺CD4⁺ T cells in the cLP, MLN and spleen of Lys-treated *Gpr43*^{-/-} mice, similar to the results of Lys administration in the WT control group (Fig. 4D-F in this PBP letter). Moreover, propionate administration in

Ido1^{-/-} mice caused significantly upregulated CD25⁺Foxp3⁺Tregs and downregulated IL17⁺CD4⁺ T cells in the cLP, MLN and spleen to a similar extent as propionate-treated WT mice (Fig. 4D-F in this PBP letter). As expected, Lys administration failed to elevate CD25⁺Foxp3⁺Tregs or lower IL17⁺CD4⁺ T cells in all three tissues in *Ido1*^{-/-} mice, while propionate treatment neither increased CD25⁺Foxp3⁺Tregs nor decreased IL17⁺CD4⁺ T cells in all three tissues in *Gpr43*^{-/-} mice compared with their respective Veh controls (Fig. 4D-F in this PBP letter). Altogether, the results from our *in vitro* and *in vivo* studies suggest that the propionate-mediated GPR43 signaling and Lys-mediated AhR-IDO1-Kyn axis are most likely not correlated with each other.

Figure 4. Propionate-mediated GPR43 signaling and the Lys-mediated AhR-IDO1-Kyn axis are most likely unrelated. (A) Expression of *Ahr* and *Ahr* target genes in BMDCs extracted from WT or *Ido1*^{-/-} mice at 6 h post-Lys treatment. (B) Expression of Lys transporters SLC7A1 and SLC7A2, and Trp transporters SLC7A5 and SLC3A2 in Lys-treated BMDCs extracted from WT or *Ido1*^{-/-} mice. (C) Naïve CD4⁺ T cells isolated from WT or *Ido1*^{-/-} mice were stimulated with immobilized anti-CD3 and soluble anti-CD28 monoclonal antibodies in the absence (untreated control; NC) or presence of propionate.

Groups of WT B6, *Gpr43*^{-/-} and *Ido1*^{-/-} mice were treated with 150 mM propionate for 3 weeks or 20 mg/kg Lys for 5 days prior to DSS treatment. At D7 post-DSS administration, CD25⁺Foxp3⁺Tregs and IL-17⁺ CD4⁺ T cells in cLP (D), MLN (E) and spleen (F) were determined by flow cytometry.

Specific Comments

Line 146-148: The authors state that Dub-treated mice exhibit less injury than Akk group, but no statistical difference between the two groups is shown in Figure S2K-N. Please explain this inconsistency.

> We apologize for the mistaken data interpretation and have changed the statement to “WT mice gavaged with Dub for 5 consecutive days starting from day 3 post-DSS administration (Fig. S2J) exhibited longer colon length (Fig. S2K-L) and improved histopathology (Fig. S2M-N) compared with mice receiving EF or untreated controls” (Line 143-146).

Line 160-163: Information about the weight loss of Abx-Dub, Abx-Akk and Abx-EF mice is lacking in Figure S3J.

> We now have provided the missing results in Fig. S3K accordingly.

Line 186-188: Please add relevant references.

> We have added the relevant references.

Line 226-227: According to Figure S6A, there is no statistical difference in butyrate or Prop levels between Dub and Akk mice. Please modify the inconsistent description in the text.

> We apologize for the mistaken data interpretation and have changed the statement to “Indeed, there were significantly higher concentrations of acetate, propionate (Prop) and butyrate in the colon samples of Dub mice compared with Akk, EF groups or the Veh controls (Fig. S6A). The same patterns were observed in fecal samples except that the concentrations of Prop and butyrate were comparable between Dub- and Akk-colonized mice (Fig. S6B).” (Line 225-229).

Line 279-281: Minor changes in protein levels, especially at D1 and D5 (Figure 4H), were not sufficient to make the claim that Dub treatment markedly increased IDO1. Better images plus quantitative scores would be needed to make this claim. Additionally, why does the IDO expression in Veh mice increase over time?

> We have now added quantitative scores to this WB image to address the reviewer’s concern (Fig. S7H). It is very reasonable that IDO1 expression increased during the course of DSS-induced colitis in Veh mice because inflammatory stimuli like IFN- γ can easily activate IDO1 both locally and systemically, driving Trp metabolism to the Kyn pathway to produce more Kyn in order to mediate inflammation remission via Treg induction. This explanation is supported by higher IDO1 activity both in the TNBS-induced colitis mouse model (Takamatsu et al., *J. Immunol.*, 2013) and in active vs nonactive IBD patients (Nikolaus et al., *Gastroenterology*, 2017), reflecting activation of the immune

system and initiation of inflammatory remission.

Line 299: Figure 4L showed the *Ido1* levels in cLPMCs, but the figure legend in Line 619 claims that it is colonic expression.

> We apologize for the inconsistent representation and have modified the description in the figure legend of Fig. 4L accordingly.

Line 369-370: Figure 5J shows that Treg cells are decreased in MLNs from Lys mice, which is inconsistent with the content in the text.

> We really appreciate that the reviewer has pointed out this mistake to us. We have corrected the content in the text to “As expected, Lys administration to GF mice (Fig. S9A) caused higher *Ido1* expression in cLP (Fig. 5M), elevated serum Kyn concentration (Fig. S9B), upregulated CD25⁺Foxp3⁺Tregs and downregulated IL17⁺CD4⁺ T cells in the spleen and cLP (Fig. 5N-O and Fig. S9C), but unexpectedly downregulated CD25⁺Foxp3⁺Tregs in the MLN (Fig. 5N, left panel).” (Line 374-378).

Line 493-495: Please add relevant references.

> We have added the relevant references to the sentence.

Line 704: “Dub” is misspelled as “Dab”.

> We have corrected the typo.

Figure 1I: Similar colors make it difficult to distinguish D7 Neo and D0 Abx, D7 Met and D0 Neo. Consider using different symbols with the same color to represent D0 and D7 respectively.

> We have chosen a more distinguishable color for different groups to avoid confusion.

Figure 5H: For the clearer presentation of data to the readers, authors should directly label the grouping information in this image.

> We now have directly labelled the grouping information in Fig. 5L.

Figure 5R, Figure S2E: Please provide scale bars.

> We now have added scale bars in Fig. 6D and Fig. S2E.

Line 320: Fig. 4A should be changed to Fig. 5A.

> We have looked at the concentrations of the five metabolites chosen from the LC-MS/MS analysis of supernatant of *D. newyorkensis* cultures (Fig. 5A) in the untargeted metabolomic analysis of the colon samples of *D. newyorkensis*-colonized mice (Fig. 4A) and found that these metabolites were also enriched following *D. newyorkensis* colonization. Thus, the description in the text is correct.

All Figures and Supplementary Figures: In the qRT-PCR analysis, the data from normalized controls is recommended to be included in the images for an easy understanding.

> We have added a dashed line at 1 in each qRT-PCR image for the better understanding that data above 1 represents upregulated expression, whereas data below 1 means downregulated expression between treatments and mock group. Hopefully this modification will ease the reviewer's concern.

Reviewer #2 (Remarks to the Author):

The authors investigated the role of commensal bacteria *Dubosiella newyorkensis* and its human equivalent, *Clostridium innocuum*, (as a last figure in the manuscript) in alleviating colitis in mice. These bacteria enhance immune tolerance by balancing Treg and Th17 responses and repairing mucosal injuries. The key mechanism involves the production of short-chain fatty acids, particularly propionate and L-Lysine. L-Lysine activates a novel metabolic circuit in dendritic cells, promoting tryptophan catabolism to kynurenine via indoleamine 2,3-dioxygenase 1 (IDO1) in an aryl hydrocarbon receptor (AhR)-dependent manner. This metabolic pathway offers a potential therapeutic target for inflammatory bowel diseases.

The study is a comprehensive yet concise exploration covering various facets, from microbial metabolism to immunoregulation. By employing multiple mouse models and delving into molecular mechanisms, it offers a complete understanding of the immunomodulatory role of these bacteria. Importantly, it identifies a potential metabolic circuit as a potential therapeutic target.

As I said, I find the study to be largely comprehensive, with minimal room for additional commentary on my part. However, I do have a few targeted inquiries that, if addressed by the authors in the manuscript, could serve to further refine specific figures without altering the overarching message of the research. I will go through my suggestions according to

the line number in the manuscript.

- Figure 1H: Is it possible to represent this data in a manner that is quantitatively measurable?

> If the reviewer refers to the histopathological data represented by H.E. staining images, we did have a scoring system to quantitate the tissue histopathological changes mentioned in the 'Materials and Method' section as previously described in detail (Allen et al., Immunity, 2012). The precise histological score of H.E. images of Fig. 1H was represented as columns in Fig. 1G (The histological score of H. E staining was shown alongside with scanned images in this way throughout the entire paper). If the reviewer refers to the IFA images showing the infiltration of IL-6-secreting macrophages, we do not have a quantitative measurement for the signals, however we did test *Il6* expression at both the transcriptional (Fig. 1D, middle panel) and translational level (Fig. 1E) in the colon as supportive data.

- It is difficult to understand how many times the authors repeated the experiments. I haven't pick up this information. Can you please add this into the figure legends? Moreover, it looks like it is only one experiment in many cases and if that's so, please change the SEM to standard deviation.

> We apologize for missing this important information in the figure legends. We now have provided the number of experimental repeats in each figure legend for the reviewer's consideration.

- Figure 1I: It looks like one ellipse is missing. Is that right? Or it is just color code issue that I cannot see it. Can you pick more distinguishable color on this panel?

> We have chosen a more distinguishable color for the different groups to avoid confusion.

- It will be nice to see how does microbiota look like after FMT in panel I-K in Figure 1?

> We now have included in the figures the microbiota composition of Abx mice that received FMT from Neo-treated mice (Fig. 1I-K), following the reviewer's consideration.

The PCoA data suggested that samples from Abx-FMT(N) mice were clustered with groups of D0. Veh and D7. Neo (Fig. 5A in this PBP letter). Further, the relative abundance of *Akkermansia* and *Dubosiella* was comparable between Neo and Abx-FMT(N) mice at D7 (Fig. 5B and C in this PBP letter).

Figure 5. Microbiota composition of Abx mice receiving FMT from Neo mice at day 7 post-DSS administration. Principal coordinate analysis (PCoA) (A) and heatmap (B) of the relative abundance of bacteria in fecal samples from mice treated with Abx or single antibiotics or receiving FMT(N) at D0 and D7 post-DSS treatment. (C) Relative abundance of *Dubosiella* and *Akkemansia* at D0 and D7 among different groups.

- In line 121, only a single publication is cited to support the protective role of certain bacteria. Could you please broaden this section by including additional relevant studies? Providing a more extensive citation list would guide readers towards the appropriate literature for further investigation.

> We have added two more references at this statement according to the reviewer's suggestion. Thanks for pointing this out to us.

- In line 133, ... even greater probiotic effect than Akk... is written. what do you mean by probiotic effect? Explain a bit further.

> We intended to say that Dub exerted an even greater protective effect against DSS-induced colitis than Akk. We have now changed the statement to 'Dub colonization exerted an even greater protective effect against DSS-induced colitis than Akk' accordingly (Line 128-129).

- The authors have demonstrated that Akkermansia exhibits comparable effects in certain aspects of the study, as evidenced in Figure 2 and Supplementary Figure S2. Given the

overlapping similarities in results, it would enhance the cohesiveness of the presentation to standardize the figures where Akkermansia data is included. For instance, how does the results look with Akkermansia when you perform TNBS model of colitis?

> Since *Akkermansia muciniphila* (*A. muciniphila*) plays an important role in attenuating IBD (Zheng et al., *Frontiers in Immunology*, 2023), we used it as a positive control to evaluate the general probiotic effect exerted by *D. newyorkensis* in mitigating DSS-induced colitis. After we determined that *D. newyorkensis* ameliorates DSS-induced colitis by rebalancing of Treg/Th17 responses, we focused on the mechanism by which *D. newyorkensis* modulates Treg-dependent immunosuppression and stopped including *A. muciniphila* in parallel in the key experiments from that point forward. However, to address the reviewer's concern, we repeated the protection study in the TNBS model of colitis with *A. muciniphila* using the same protocol described in Fig. S3A of our revised paper. Indeed, *A. muciniphila* colonization also mitigated disease phenotypes (Fig. 5A-B in this PBP letter), histopathological changes (Fig. 5C in this PBP letter) and proinflammatory cytokine expression (Fig. 5D in this PBP letter), and improved mucosal barrier integrity (Fig. 5E in this PBP letter).

Figure 6. Oral administration of *A. muciniphila* ameliorates TNBS-induced colitis. Groups of conventional WT B6 mice (n=6) were gavaged with 10^9 CFU of *A. muciniphila* (Akk) or vehicle (Veh) and treated with TNBS. Measurement of colon length (A and B) and histopathological evaluation (C) by HE staining was performed. The level of *Il1b*, *Il6* and *Tnfa* in the colonic lamina propria cells (cLPs) (D) and the expression of *ZO-1*, *OCLN* and *Muc2* in colonic intestinal epithelial cells (cIECs) (E) were determined by qRT-PCR.

- Is the same set of genes upregulated in Akk treated group if one performed the experiment in Figure 3A?

> Actually, we performed the transcriptomic analysis of colon samples from conventional B6 mice colonized with *D. newyorkensis* or *A. muciniphila* and found fundamental differences in transcriptional profiles between *D. newyorkensis*-colonized versus *A. muciniphila*-colonized mice. We now provide the heatmap of differentially expressed genes (DEGs) between the two bacterially colonized groups in response to the reviewer's concern.

Figure 7. Heat map showing mRNA expression determined by bulk RNA-Seq in colon samples from vehicle (Veh)-treated (n=3), *D. newyorkensis* (Dub)- or *A. muciniphila* (Akk)-colonized wild-type C57BL/6J mice (n=5) at D7 post-DSS treatment.

- How does the phenotype look with monocolonization with *Akkermansia*?

> If the reviewer refers to the protective phenotype of monocolonization with *A. muciniphila* in germ-free (GF) mice, we did not perform the protection study in a GF mice model. Nevertheless, we did colonize the antibiotics-treated (Abx) mice with *A. muciniphila* and showed that *A. muciniphila* colonization in Abx mice led to decreased loss of body weight (Fig. S3K), longer colon length (Fig. 2E and Fig. S3L), lower histopathological score (Fig. 2F-G) and less macrophage infiltration (Fig. 2G) in the colon, indicating that *A. muciniphila* conferred protection against DSS-induced colitis even less effectively than *D. newyorkensis*.

-In Figure 4A, does the displayed data encompass all statistically significant findings? If so, could you please incorporate the corresponding statistical annotations? Additionally, could you elaborate on the distinctions between the various groups represented?

> The heatmap of untargeted metabolomic analysis displayed the top 35 most abundant microbial metabolites detected in the colon samples of various bacteria-colonized mice. Among them, 9 metabolites (glutaryl carnitine, 9-pentadecenoic acid, docosapentaenoic acid 22n-6, L-lactic acid, stearyl carnitine, L-cystine, L-2-aminobutyric acid, N-methyl-L-glutamic acid and eicosapentaenoic acid (EPA) were statistically significantly upregulated or downregulated in the colon of Dub-colonized mice vs noncolonized mice. To address the reviewer's concern, we have incorporated the corresponding statistical annotations of these metabolites in the heatmap and described the distinctions between the various groups in the text (Line 264-266).

Reviewer #3 (Remarks to the Author):

In this manuscript, Zhang et al. discovered *D. newyorkensis*, an underappreciated murine commensal bacterium that produces short-chain fatty acids and can balance intestinal Treg/Th17 responses, thereby repairing mucosal barrier damage and mitigating DSS-induced colitis in mice. The authors demonstrated that *D. newyorkensis* and its human homologue *C. innocuum* produce L-Lysine to enhance tryptophan metabolism in human and mouse dendritic cells, which subsequently induced kynurenine production and Treg-mediated immunosuppression. Mechanistically, it appears that GPR43, IDO1, and the aryl hydrocarbon receptor were involved in *D. newyorkensis*'s immunoregulatory action.

Overall, the results of this study provide an important mechanistic understanding of how commensal bacteria create an immunosuppressive milieu through the secretion of immunomodulatory compounds. The mechanistic model builds with suitable validation and controls and is generally logical. The authors' ability to begin with a thorough functional analysis of a recently discovered murine commensal and end up with a human homolog that has a comparable biological effect is quite impressive. A couple of things are in need of more clarification or experimental evidence:

1. The authors have demonstrated that the relative abundance of the *Dubosiella* genus was significantly higher at D0 and D7 post-DSS treatment (Fig. 1K). How about the

absolute abundance of *D. newyorkensis*? The authors need to ascertain whether this data aligns with the relative abundance of *Dubosiella*.

> We have now analyzed the absolute abundance of *D. newyorkensis* in Veh, Abx, and single-antibiotic-treated mice at D7 via qPCR and found significantly higher bacterial copies of *D. newyorkensis* in Neo- and Metro-treated mice vs the other single-antibiotic-treated and Abx mice. These results are in line with the relative abundance of *Dubosiella* observed in 16S rRNA gene sequencing analyses (Fig. 8 in this PBP letter).

Figure 8. 16S rDNA copies per fecal pellet in mice treated with different single antibiotics (Amp, Van, Neo or Metro) or in combination (Abx) at day 7 post-DSS administration.

2. Please also provide the absolute abundance of *D. newyorkensis* in the fecal samples collected from mice that underwent fecal transplantation or antibiotic treatment (Fig. 1A).

> As suggested, we have now provided the absolute abundance of *D. newyorkensis* in fecal samples collected from Veh, Abx, Abx-FMT and Neo-treated mice. As shown in Figure 9 in this PBP letter, Abx-FMT mice displayed significantly increased bacterial copies of *D. newyorkensis* compared to Abx mice, but not to the same level as the Veh

control.

Figure 9. 16S rDNA copies per fecal pellet in mice treated with antibiotic cocktail (Abx), Abx mice given fecal microbiota transplantation (FMT) from Neo-treated mice [Abx-FMT(N)], or mice treated with Neo at day 7 post-DSS administration.

3. Given that the bacterium produces high levels of SCFAs, particularly propionate (Fig. 3G and Fig. S6A), what pathway do the authors think *D. newyorkensis* uses to produce propionate?

> To predict the pathways that Dub might use to produce propionate, we aligned the putative amino acid sequence of Dub with the KEGG Orthology database using the DIAMOND program (v2.0.11.149). Our analysis revealed that Dub encodes multiple key enzymes involved in propionate synthesis (Figure 10). However, we did not identify any complete pathways capable of de novo generation of propionic acid from substrates such as lactate, succinate, or 1,2-propanediol. Given the bacterial strain's distinct metabolic characteristics in lactate production, we speculate that the most likely pathway for propionate production in Dub is via lactic acid. This pathway appears to be missing only one enzyme (EC: 3.2.1.54). The absence of a complete pathway may be attributed to

gaps in the genome sequence or possibly to a novel biosynthetic mechanism that has not yet been identified, which warrants further investigation.

Figure 10. Reconstruction of propionate-associated metabolic pathways, with reference to KEGG map00640. The blue nodes represent various compounds. Arrowed lines indicate the direction of chemical reactions. Lines highlighted in red depict the most likely pathway for propionate synthesis we have predicted. The enzyme names colored in red and green indicate their presence or absence, respectively, in the Dub genome. The accompanying bar plots display the expression levels of several metabolites in Dub and the control group, as detected by LC-MS technology.

4. Is there any experimental proof for the initial choice of propionate as the SCFA representative, without taking acetate or butyrate into account when examining *D. newyorkensis*'s immunosuppressive effects?

> To address the reviewer's concern, we administered acetate, propionate, butyrate and the mixture of three SCFAs to groups of WT B6 mice for 3 weeks (Fig. 11A-B in this PBP letter), then treated with DSS and assessed the disease phenotype, histological changes, mucosal barrier integrity and proinflammatory cytokine expression among the different

groups. It was demonstrated that propionate administration conferred protection against DSS-induced colitis comparably to acetate in ameliorating colon shortening (Fig. 11C-D in this PBP letter) and pathological changes (Fig. 11E in this PBP letter), whereas propionate exerted a better protective effect than butyrate or the SCFA mixture taking into account these parameters. Furthermore, propionate treatment exhibited the best performance in downregulating *Ilf6* expression among all SCFA treatments, while the level of *ZO-1*, *Muc2*, *Ilf1b* and *Tnfa* transcripts was equivalent among all three SCFA-treated groups and the mixture (Fig. 11F-G in this PBP letter). Thus, we chose propionate as the SCFA representative based on the fact that it exerted the best protective effect in mitigating DSS-induced colitis in most of the in vivo parameters.

Figure 11. SCFAs protect mice from DSS-induced colitis. Conventional WT mice (n=4-6) were administered either sodium acetate (Ace. 150 mM), sodium propionate (Prop. 150 mM), sodium butyrate (But. 100 mM) or a mixture of SCFAs (67.5 mM Ace., 40 mM But., 25.9 mM Prop.) in the drinking water for 3 weeks before DSS exposure (water solutions were prepared and changed every 3 days) (A). Fecal SCFA concentration of treated animals was determined by GC/MS (B). Measurement of colon length (C and D) and histopathological evaluation (E) by HE staining was performed. The level of *I11b*, *I16* and *Tnfa* in the colonic lamina propria cells (cLPs) (F) and the expression of *ZO-1*, *OCN* and *Muc2* in colonic intestinal epithelial cells (cIECs) (G) was determined by qRT-PCR.

Minor points:

1. I've noticed that lactate is one of the five most abundant metabolites derived from the cultured supernatant of *D. newyorkensis*. Also, giving lactate to the animals receiving DSS treatment reduced intestinal inflammation, tight junction damage, and disease phenotypes (Fig. S8E). Why was L-lysine chosen as the target metabolite in the following study rather than lactate?

> Indeed, administration of lactate in WT conventional B6 mice led to reduced intestinal inflammation, tight junction damage and disease phenotypes upon DSS treatment (Fig. S8C). This finding was consistent with one previously published study that symbiont-derived lactate is sensed by GPR81 on Paneth and stromal cells to promote regeneration in a Wnt3/ β -catenin-dependent manner. Lactate administration has also been shown to be protective against gut damage provoked by radiation and

chemotherapy (Lee et al., Cell Host Microbe, 2018). Furthermore, a recent study conducted by Gu et al. suggested that lactate improves Treg cell stability and function in maintaining the immunosuppressive tumor microenvironment by modulating MOESIN lactylation and enhancing TGF- β signaling (Gu et al., Cell reports, 2022). Based on the already known functions and mechanisms involved in lactate-mediated wound healing of the intestinal mucosal barrier and modulation of immunosuppressive Tregs, we chose to focus on the potential role of an underappreciated metabolite, Lys, in modulating Treg responses in colitis.

2. There was too much data included in Fig. 5. It may be helpful to split the figure into two for easier understanding.

> We have modified Fig. 5 and the text according to the reviewer's suggestion.

3. Has the purity of the primary DCs isolated from human PBMCs been validated by the authors?

> We now have provided the purity of primary DCs extracted from human PBMCs as following for the reviewer's consideration:

Figure 12. Representative flow cytometry plots showing CD11c⁺ DCs extracted from human PBMCs.

4. The color scheme of Fig. 5V should be matched to Fig. 5Y and 5Z.

> The figure panels have been corrected in the revised version.

REVIEWERS' COMMENTS

Reviewer #1 (Remarks to the Author):

The revised manuscript has basically solved the issues I am concerned about and can be published in the current version.

Reviewer #2 (Remarks to the Author):

I would like to thank the authors for making efforts to reply our questions. I have no further comments.

Reviewer #3 (Remarks to the Author):

I thank the authors for sufficiently addressing my comments.